# Towards the systematic reconnaissance of seismic signals from glaciers and ice sheets - Part A: Event detection for cryoseismology

Rebecca B. Latto[1], Ross J. Turner[1], Anya M. Reading[1], and J. Paul Winberry[2]

[1]School of Natural Sciences (Physics), University of Tasmania, Private Bag 37, Hobart, 7001, Australia
[2]Department of Geological Sciences, Central Washington University, Ellensburg, WA, USA

**Correspondence:** Rebecca B. Latto (beccablatto@gmail.com)

**Abstract.** Cryoseismology is a powerful toolset for progressing the understanding of the structure and dynamics of glaciers and ice sheets. It can enable the detection of hidden processes such as brittle fracture, basal sliding, transient hydrological processes, and calving. Addressing the challenge of detecting signals from many different processes we present a novel approach for the semi-automated detection of events and event-like noise, which is well-suited for use as Part A of a workflow where unsupervised machine learning will be used as Part B to facilitate the main reconnaissance of diverse detected event types. Implemented in the open source and widely used ObsPy python package, the multi-STA/LTA algorithm constructs a hybrid characteristic function from a set of sta, lta pairs. We apply the algorithm to data from a seismic array deployed on the Whillans Ice Stream (WIS) in West Antarctica (austral summer 2010–2011), to form a 'catch-all' catalogue of events and event-like noise. The new algorithm compares favorably with standard approaches, yielding a diversity of seismic events, including all previously identified stick-slip events (Pratt et al., 2014), teleseisms, and other noise-type signals. In terms of a catalogue overview, we investigate a partial association of seismicity with the tidal cycle, and a slight association with ice temperature changes of the Antarctic summer. The new algorithm and workflow will assist in: the comparison of different glacier environments using seismology, the identification of process change over time, and the targeting of possible following high-resolution studies.

## 1 Introduction

Seismic event detection is an important initial processing step in the analysis of signals from a seismic network. Using human analysts or automated techniques with analyst review, the objective is to form an event catalogue representative of the seismicity during a time period. Event catalogues have value as they may be used for seismicity studies or as a working database for further scientific investigation. For these purposes, catalogues aid the comparison between localities, and improve the detection of change over time. Equivalent to human 'picking', automated techniques detect seismic signals above background noise and calculate the magnitude and other quantitative characteristics of events using computational means. With regard to

earthquake seismology, automated event detection techniques such as STA/LTA ('short-term average over long-term average') and correlation-type, have been used with increasing success and continue to develop (Bergen and Beroza, 2019). These techniques are suitable for the seismological analysis of specific examples of the smaller and more varied signals from volcanoes, landslides, mine activity and glacier processes.

In contrast to targeting the event detection to a specific signal or event type, we take an alternative approach to the detection process, intending it to be 'catch-all' such that it includes both events and event-like noise. We use the term 'event' broadly to include impulsive signals, and waveform changes (such as an amplitude increase or frequency content change) with a less distinct onset. In some glacier environments, event-like noise is of as much interest as impulsive cryoseismic events, as both signal types yield insight into glacier and/or ice shelf processes. The workflow that we develop through this contribution thus aims to capture the wide variety of seismic events and event-like noise present in a glacier environment and any adjacent ice shelf (Part A) and subsequently undertake a reconnaissance of event types using unsupervised machine learning (Part B, Latto et al., 2024). This workflow aims to enable the reconnaissance of ice-covered environments, such as outlet glaciers of ice sheets, some of which supply ice shelves. It provides a consistent and repeatable approach that will work with a modest number of stations deployed over a wide, remote area to provide an initial appraisal of seismicity across a given region. Such a reconnaissance could facilitate either (1) a comparison of the processes active in different locations; and/or (2) the monitoring of glacier processes over time; and/or (3) the targeting of following high-resolution studies.

The ObsPy Python project (Beyreuther et al., 2010) is a software package, widely used at the time of writing, for observational seismology including the implementation of the STA/LTA algorithm (or STA/LTA for short) and other data analysis tools in seismology research. As a framework for processing observational seismological data, ObsPy provides a community-wide resource, and we have drawn upon the nomenclature used therein unless otherwise stated. The STA/LTA algorithm calculates average values of absolute waveform amplitude in two time windows, one short and one long, and compares their ratio to a threshold value for detection (Allen, 1982; Trnkoczy, 2009). An event is triggered (i.e. an event arrival time is picked) at the point where the ratio rises above the trigger threshold value and is detriggered (i.e. an event stop time is picked) once it falls below the detrigger threshold value. When discussing the STA/LTA algorithm, we are explicitly referring to the recursive STA/LTA algorithm, which may be regarded as a current standard approach of this type. The recursive STA/LTA algorithm improves upon the classic algorithm as it reduces memory usage, yields a decaying exponential impulse response instead of a rectangular one, and limits shadow zones. Shadow zones occur when short event bursts (transients) in the STA cause a large LTA, and the recursive algorithm limits the dominating effect that such transients could have on successive event detections. All together, this produces a more efficient and smoother result (Withers et al., 1998). A shortcoming of the STA/LTA algorithm is that it is sensitive to changing noise levels. In environmental seismology studies, this leads to missed detections because the signals of interest have low signal-to-noise ratios and/or are diverse with regard to maximum amplitude and time scales (Vaezi and Van der Baan, 2015).

In comparison to STA/LTA, correlation-type algorithms assume similar events will occur within a catalogue so can be less prone to missed detections of specified events because they typically include some type of template matching (Withers et al., 1998). This involves computing a normalized correlation coefficient of a template for a previously-known waveform and

searching a dataset for similar waveforms (Anstey, 1966). Under the umbrella of correlation-type algorithms, seismic studies have employed stacking algorithms (Grigoli et al., 2016) and artificial neural networks (Wang and Teng, 1995, 1997; Valentine and Trampert, 2012). Other advanced techniques include assigning nonlinear filters (Perol et al., 2018) and the computationally efficient similarity search (Yoon et al., 2015). Despite successful applications in earthquake seismology, correlation-type algorithms can lead to missed detections in environmental seismology because of the varied and previously undetermined waveforms of the signals of interest.

Since STA/LTA and correlation-type algorithms have enjoyed only limited success when applied to environmental seismology, there is no community-wide consensus on a best event detection technique for generating event catalogues for cryoseismic signals. In fact, the many different methods that have been applied demonstrate the experimental and individualized approach to detection in cryoseismology (Podolskiy and Walter, 2016a; Aster and Winberry, 2017). The most basic method is manual detection (Pomeroy et al., 2013; Pratt et al., 2014; Barcheck et al., 2018), which avoids missed and false detections associated with automated event detection. However, visually picking event arrival times in continuous cryoseismic data is time consuming and analyst dependent. In terms of automated detection, many studies use the STA/LTA algorithm, which can be restrictive (i.e. only recognizes a single set of parameters) and so is difficult to apply across the diversity of signals that can come from various glacier environments. Therefore, several cryoseismic and environmental seismic studies have adapted STA/LTA to certain applications. For example, Bassis et al. (2007) design a variation of STA/LTA based on the detection requirements of the seismometers in their study. In this case, a small sample of manually detected events is used to optimize parameters. Similarly, other hybridized approaches to STA/LTA are presented in Cichowicz (1993) and Lois et al. (2013) for the purpose of more accurate detection in diverse signal-and-noise environments, like those of microseismic networks. Accounting for the possibility of missed detections, Roux et al. (2008) rely on post-processing events detected by STA/LTA (e.g. extending an event's length, accounting for potential low amplitude events that can be left undetected). Minowa et al. (2019) apply STA/LTA to data filtered by two separate frequency bands, such that parameters can be adjusted for event types characterized by different frequencies. Ultimately, most studies determine the parameters by trial and error (e.g. Lombardi et al., 2019).

Alternatives to the amplitude dependent STA/LTA are the use of other measures of an event, such as kurtosis, i.e. quantifying deviations from a standard distribution of waveform amplitudes (McBrearty et al., 2020), and spectrograms for frequency-related thresholds for detection (Helmstetter et al., 2015). QuakeMigrate, an advanced extension of the STA/LTA algorithm and waveform stacking, relies on a coalescence (i.e. joint) energy approach that quantifies the cumulative seismic energy recorded by a network of stations (Smith et al., 2020). While QuakeMigrate, and spectral-based methods, have been shown to be effective for detecting basal icequakes, they are less effective at capturing longer tremors (Hudson et al., 2019). Some studies also use STA/LTA and correlation-type methods together (e.g. Walter et al., 2008; Allstadt and Malone, 2014; Köhler et al., 2015), or just use correlation-type (Mikesell et al., 2012), when there is a reference waveform available for template matching. An example of a more advanced technique on glaciers is the use of hidden Markov models, which provide rapid, precise detection via learning, but rely on labelled training data (Hammer et al., 2015). Where high-resolution sensor coverage is desirable and possible, source locations and glacier processes may be determined directly (e.g. Nanni et al., 2022, make use

of a dense, ~800 m aperture array), with the reconnaissance-level approaches that we describe enabling the targeting of such detailed studies.

The wide variety of techniques for the detection of icequakes highlights the extent of analytical challenges in event-based cryoseismology. Where the area of interest is an ice stream or other ice sheet outlet glacier, the challenge is increased by the remote location together with the need to undertake a reconnaissance across a relatively large area. The diversity of event types in glacier environments therefore suggests the need for a workflow comprising a (Part A) 'catch-all' algorithm that can automatically capture heterogeneous seismicity characteristics. Given the advent of machine learning research, a rapid and broad event detection method is a critical tool for preparing datasets for subsequent (Part B) semi-automated reconnaissance. In terms of any type of analysis, a consistent approach from glacier to glacier will be particularly useful. The high potential utility of comparable event catalogues being generated from different seismic deployments provides motivation to develop a generalized approach for event detection and catalogue compilation for cryoseismology data.

In this contribution, as Part A of the workflow proposed above, we outline the development and testing of a novel event detection method, the multi-STA/LTA algorithm, that uses a set of sta, lta pairs to optimize the detection of diverse cryogenic signals. We form a synthetic set of test waveforms, using a Monte Carlo approach, and then perform a grid search to inform our choice of parameters for the subsequent application of the algorithm. We also compare the event detection performance of the multi-STA-LTA algorithm with the recursive STA/LTA method. In a subsequent section, we apply the multi-STA/LTA algorithm to a dataset from the Whillans Ice Stream to form an event catalogue, and demonstrate how our semi-automated event detections are substantiated by a number of visual-based event detections. Our broad use of the term 'event' includes both impulsive signals and waveform changes with a less distinct onset. The new event catalogue may be considered sufficiently comprehensive to allow for an appraisal of the recorded glacier seismicity, and lends itself to subsequent analysis using unsupervised maching learning (Part B, Latto et al., 2024). Finally, we discuss the limitations and utility of this methodology in cryoseismology, and other environmental seismology applications.

## 2   multi-STA/LTA algorithm

In this section, we develop the multi-STA/LTA algorithm to detect events with a range of durations and maximum amplitudes. This algorithm is based on the principles of the established STA/LTA procedure (Trnkoczy, 2009). We use a set of simulated waveforms for algorithm testing using a Monte Carlo approach, and carry out a fine grid search to inform the choice of multi-STA/LTA algorithm parameters, defined in Table 1. The relevant functions of the ObsPy package use nomenclature including the use of 'sta' (lower case) as a duration for a short time window and correspondingly 'lta' refers to the span of a long time window, with both given in seconds. The short and long time windows are used to calculate the average amplitudes within these window spans. We explicitly use the uppercase 'STA' and 'LTA' as abbreviations for the short- and long-term averages. STA/LTA is thus a ratio of the averaged amplitudes within these short and long time windows.

**Table 1.** Parameter definitions for the multi-STA/LTA algorithm, based on their usage in ObsPy, see main text for clarification. The right-hand column shows the ranges of the parameter values sampled in a fine-grid search for best parameter value choice (Sect. 5), the step size within the given ranges is uniformly distributed in $\log_{10}$ space.

| Parameter name (units) | Definition | Range in parameter search |
|---|---|---|
| sta (seconds) | span of the minimum short time window | 0.001–100s |
| lta (seconds) | span of the minimum long time window | 1–100s |
| $\Delta_{sta}$ (unitless) | multiplier of sta (used to compute the span of the maximum short time window) | 10–1000 |
| $\Delta_{lta}$ (unitless) | multiplier of lta (used to compute the span of the maximum long time window) | 10–1000 |
| $\epsilon$ (unitless) | the target ratio between sta, lta pairs (used as a tolerance value to limit the number of time windows that are generated to ensure computational efficiency) | 1.78–100 |

## 2.1 Algorithm description

The foundation of the multi-STA/LTA algorithm is the formation of a hybrid characteristic function (CF) for optimized event
detection. In statistics, a CF often represents a probability distribution of maximum eigenvalues (Feuerverger and Mureika, 1977). When applied in this context in seismology, the CF is a non-linear transform of a seismogram showing a probability distribution of likelihood for event detection. Physically based, it quantifies changes in the energy in a waveform's direction of displacement.

The multi-STA/LTA algorithm takes the given input parameters (Table 1) and forms a hybrid CF through the following four
steps:

1. Generate a set of short- and long-time window 'sta, lta pairs'. The algorithm determines the number and spans of sta, lta pairs based on the input parameters: sta, lta, $\Delta_{sta}$ and $\Delta_{lta}$, and $\epsilon$.

2. Calculate the CF for each sta, lta pair using the recursive STA/LTA algorithm applied to an input waveform. The CF is the ratio of the absolute waveform amplitudes averaged within the short- and long-time windows.

3. Calculate a hybrid CF from the maximum values of each CF computed in Step 2 (i.e. maximum STA/LTA ratios) using Eq. (1).

4. Trigger (or continue) an event while the hybrid CF value at a given point in time is above a trigger threshold and detrigger an event when the hybrid CF falls below a detrigger threshold, yielding an event list for the desired time period.

Larger values in a CF signify a greater likelihood of an event having occurred at that time; therefore, the maximum value
of the CF at a given point in time should signify the highest likelihood of an event. The result is a hybrid CF that is tailored systematically to each event in a waveform. This contrasts to the previously defined STA/LTA approach which uses one CF computed by one short and long time window. The multi-STA/LTA algorithm thus innovates upon the recursive STA/LTA algorithm by merging multiple CFs, resulting in a hybrid CF.

The parameters of the hybrid CF used in the multi-STA/LTA algorithm, defined by the following equation, are selected to optimize the successful detection of events and to minimize false detections:

$$C_{\mathrm{multi}}(A_{\mathrm{on}}, A_{\mathrm{off}}, \mathrm{sta}, \mathrm{lta}, \Delta_{\mathrm{sta}}, \Delta_{\mathrm{lta}}, \epsilon) = \max\left\{ C_{\mathrm{recursive}}(A_{\mathrm{on}}, A_{\mathrm{off}}, \mathrm{sta} \times (\Delta_{\mathrm{sta}})^{i/(n-1)}, \mathrm{lta} \times (\Delta_{\mathrm{lta}})^{i/(n-1)}) : i = 0, \ldots, n-1 \right\},$$

(1)

where the function $C_{\mathrm{multi}}$ is the CF for the multi-STA/LTA algorithm and the function $C_{\mathrm{recursive}}$ is the CF for the recursive STA/LTA algorithm implemented per sta, lta pair. $A_{\mathrm{on}}$ and $A_{\mathrm{off}}$ are thresholds for which an event is triggered and detriggered respectively, and $n$ is the number of sta, lta pairs, defined as the smallest integer satisfying the equation:

$$n > \ln(\max(\Delta_{\mathrm{sta}}, 1/\Delta_{\mathrm{sta}}, \Delta_{\mathrm{lta}}, 1/\Delta_{\mathrm{lta}}))/\ln(\epsilon).$$

(2)

As an example, to demonstrate how the algorithm generates a set of sta, lta pairs (Step 1), using parameter values of sta = 1 second, lta = 10 seconds, $\Delta_{\mathrm{sta}} = 10$, $\Delta_{\mathrm{lta}} = 10$, and $\epsilon = 2$ as inputs to the multi-STA/LTA algorithm the following pairs are generated:

$$\{(\mathrm{sta}_0/\mathrm{lta}_0), ..., (\mathrm{sta}_i/\mathrm{lta}_i), ..., (\mathrm{sta}_{n-1}/\mathrm{lta}_{n-1})\} = \{(1/10), (2.15/21.5), (4.64/46.4), (10/100)\}.$$

(3)

The corresponding CFs (i.e. absolute amplitude ratios) will be calculated, for each pair in this set, and the maximum CF per time window will be used to form the hybrid CF (Steps 2 and 3). We illustrate the construction of the hybrid CF for this example parameter set applied to a waveform in Fig. 1.

## 2.2 Algorithm testing and optimization

We use a Monte Carlo simulation to provide a set of waveforms for algorithm development. Each waveform thus generated includes randomized background noise and two events that vary in time of occurrence, duration, amplitude, and time between the events. The range of seismic signals that we intend to capture in our simulations includes: long-period events with decaying amplitude; two events that are proximate in time with different amplitudes and durations; and low amplitude events whose structure is barely detectable above background noise. The large population of simulated events are positioned at varying temporal separations with respect to each other in each simulated waveform. The set of simulated waveforms is not intended to accomplish the difficult task of replicating the exact nature of likely cryoseismic signals, but rather to provide a working set of signals that present similar challenges to the STA/LTA algorithm, for example, events of different types being closely spaced in time.

We define two event classes and randomly sample for a uniform distribution over the variables of each class. The sample space for each class is shown in Table 2. Further details of the simulated waveforms including the equations for the two classes of simulated events, Eq. (S1a) and Eq. (S1b), and further discussion of the simulated seismic signal are described in

the Supplementary Materials (Sect. S2.2a). Following the definitions of the event classes and sample space, we use statistical analyses to guide parameter choices for the algorithm.

We find the marginal probability distribution results for sta, lta, $\Delta_{\text{sta}}$ and $\Delta_{\text{lta}}$, and $\epsilon$ (Fig. 2). Finding ambiguity in the choices of $\Delta_{\text{sta}}$ and $\Delta_{\text{lta}}$, we also look at the marginal probability distributions of the two parameters (Fig. 3). These figures are provided in the main text to support the parameter recommendations. However, full discussion of the figures and statistical methods applied are detailed in Sect. S2.2b. Limitations to these methods, as they are applied, are examined in Sect. 4.1.

**Table 2.** Sample space for the variables of each simulated event class. The values per variable are randomly sampled for a uniform distribution in $\log_{10}$ space, and are used to generate a simulated waveform containing two events as defined by Eq. (S1a) and Eq. (S1b). Further discussion of the two event classes is provided in the Supplementary Materials (Sect. S2.2a).

| Variable | Sample space for Eq. (S1a) | Sample space for Eq. (S1b) |
|---|---|---|
| Amplitude ($A$) | 1–1000 | 1–1000 |
| Duration ($T$) | 1–100 | 1–100 |
| Period component ($n$) | 1–10 | 1–10 |
| Period component ($m$) | – | 10–100 |
| Decay ($\beta$) | 1–3 | 1–3 |
| Constant ($\gamma$) | – | -1–1 |

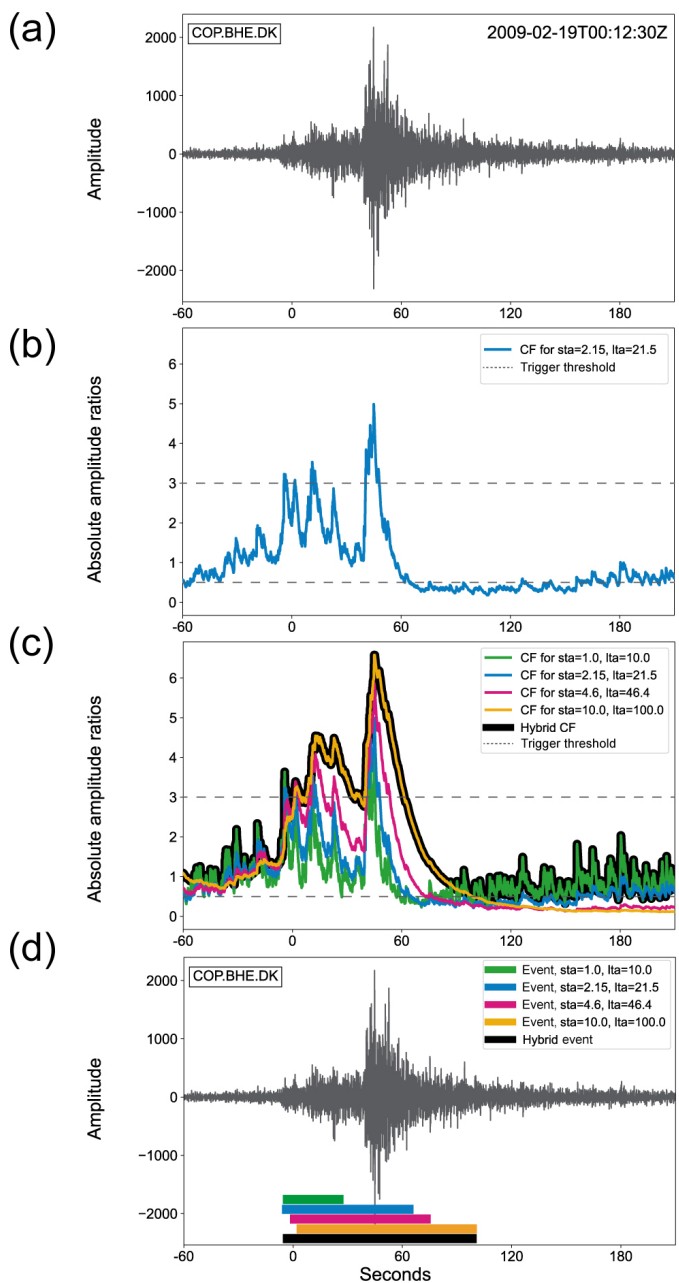

**Figure 1.** Example to illustrate the differences in event detection for a recursive vs a multi-STA/LTA approach. In the first panel (a) we provide a seismic waveform sourced from https://examples.obspy.org/ with a highpass 1 Hz filter. In (b), the CF for a single recursive sta, lta pair (sta = 2.15 seconds, lta = 21.5 seconds) whereby one set of sta, lta pairs is used to calculate the STA/LTA amplitude ratios, is provided as a point of comparison to (c), the hybrid CF (black) with the CFs for the four sets of sta, lta pairs in Eq. (3) that are generated by the multi-STA/LTA parameters sta = 1 second, lta = 10 seconds, $\Delta_{sta} = 10$, $\Delta_{lta} = 10$, and $\epsilon = 2$. Two reference lines for a typical trigger value of 3 and a detrigger value of 0.5 (grey, dashed) are overlaid in (b) and (c); these values are picked to show the best comparison between event detections, but are not subsequently applied to real data. In (d) the waveform is repeated from (a) with underlying horizontal bars showing the extents of each detected event per each CF in (b) as they meet the respective trigger and detrigger thresholds. The multi-STA/LTA detected event (black) is at the maximum extent for the set of 4 recursive STA/LTA detected events.

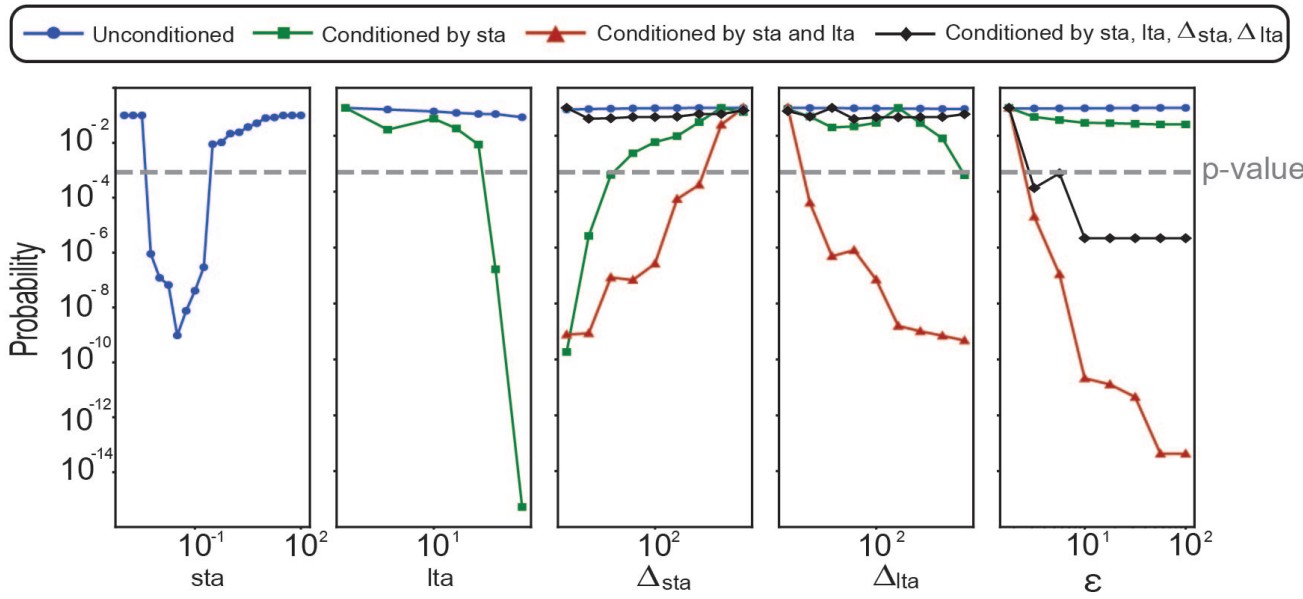

**Figure 2.** A guide to parameter choice for the multi-STA/LTA algorithm: the marginal probability distributions of sta, lta, $\Delta_{sta}$, $\Delta_{lta}$, and $\epsilon$. Each scatter plot reflects the fine grid of tested parameter values. Displayed are unconditioned values (blue circles), values conditioned by sta between 0.01 and 0.18 (green squares), values conditioned by the sta and lta at 100 (red triangles), and values conditioned by sta, lta, and the line $\log_{10}(\Delta_{lta}) = 1.24 \times \log_{10}(\Delta_{sta}) + 0.06$ (black diamonds). The threshold for statistical significance is shown at 0.005 (grey dashed line). Further discussion of the fine grid parameter search is provided in the Supplementary Materials (Sect. 2.2b).

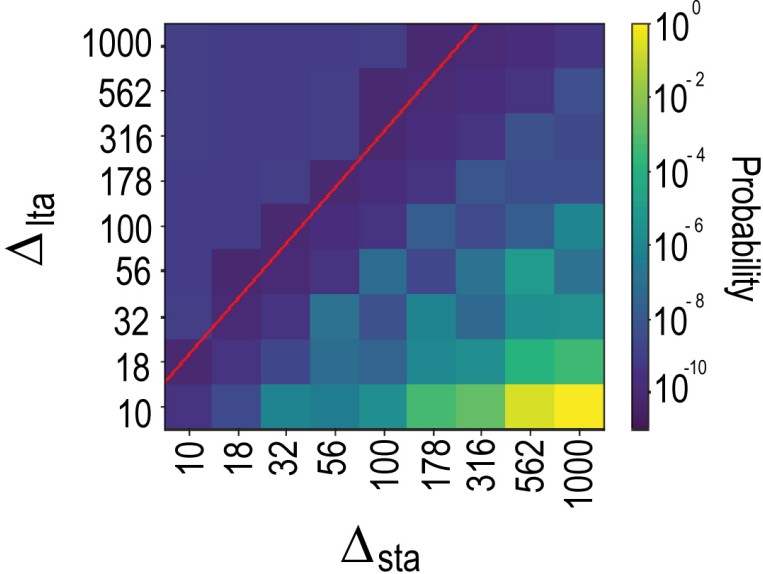

**Figure 3.** A further guide to parameter choices of $\Delta_{sta}$ and $\Delta_{lta}$: the two-dimensional marginal probability distribution for $\Delta_{sta}$ and $\Delta_{lta}$. Overlaid is the line of best fit for the points of highest success of event detection (red line: $\log_{10}(\Delta_{lta}) = 1.24 \times \log_{10}(\Delta_{sta}) + 0.06$). A low probability signifies a rejection of the null hypothesis that events and noise are equally likely to be detected, therefore, a higher success in event detection. The colorbar is scaled by the maximum probability of rejecting the null hypothesis. Further discussion of the fine grid parameter search is provided in the Supplementary Materials (Sect. 2.2b).

## 3 Whillans Ice Stream event catalogue

We now apply the multi-STA/LTA algorithm to a real cryoseismology dataset – seismic recordings from the Whillans Ice Stream (WIS; also known as Ice Stream B) – with the aim of generating an event catalogue spanning the deployment period in the 2010–2011 austral summer.

### 3.1 Data

The WIS is one of five major glaciers that discharge ice from the grounded West Antarctic Ice Sheet into the Ross Ice Shelf. It is a fast flowing ice stream at 300 m a$^{-1}$, due to its well-lubricated, deformable subglacial bed (Tulaczyk et al., 2000); however it is also decelerating at an estimated rate of 5.5 m a$^{-2}$ (Beem et al., 2014), resulting in a high deformation rate. It experiences large-scale stick-slip motion with tidal effects from the Ross Sea and Ross Ice Shelf. The WIS is therefore a locality with the potential for a wide range of cryoseismic events such as signals related to resonance in subglacial water-filled cracks (Winberry et al., 2009) and glacier earthquakes and tremors relating to the release of strain the ice-bed boundary (i.e. stick-slip; Winberry et al., 2013; Lipovsky and Dunham, 2016).

The WIS is a challenging case study for cryoseismic studies because its proximity to the Ross Ice Shelf (RIS) contributes to an especially noisy seismic wavefield. Though the RIS front is distant from the WIS deployment (up to 600 km), indirect, external sources can still be detected on the WIS (Wiens et al., 2016), therefore, seismicity recorded on the RIS and WIS could have various potential generative mechanisms. As well as the stick-slip events noted above, signals are possible from teleseismic events (Baker et al., 2021), ocean swell and gravity waves (Chen et al., 2019), surface resonance (Chaput et al., 2018), rift fracture (Olinger et al., 2019), and flexure of the frozen surface (MacAyeal et al., 2019). Environmental factors that are associated with detections of higher frequency signals are tidal stresses, changes in air temperature or insolation, and wind speed (Jenkins et al., 2021).

Seismic sensors were deployed on the WIS in West Antarctica during the 2010–2011 and 2011–2012 austral summers (Winberry et al., 2010; Pratt et al., 2014). From the 49 stations deployed, we examine data from 14 stations (station names of format: BBXX; Fig. 4, red filled circles). The BBXX seismometers are all Nanometrics Trillium 120 Sec instruments with Reftek 130 D recorders using 200 Hz sampling. Each station has continuous three component data for the time period between 2010:12:14 and 2011:01:31. Excluded are stations BB02, BB05, and BB09 due to missing components and/or incomplete data for a significant proportion of the deployment.

A common practice of the recursive STA/LTA algorithm in earthquake detection is to use only the vertical component, however, we also wish to make use of the horizontal components. This is a practical solution in case of the an instrument problem on the vertical component, and accounting for possible ray paths of seismic events to sensors. We utilize the root sum squared (Euclidean norm) of the North, East, and Vertical (Z) component amplitudes for each station; this allows weaker signals to be detected by the algorithm irrespective of component. This absolute value (no implied physical meaning) is simply a means of triggering events. Conventional frequency filtering is not applied during pre-processing in order to maintain the full range of frequencies available to be captured by the multi-STA/LTA algorithm.

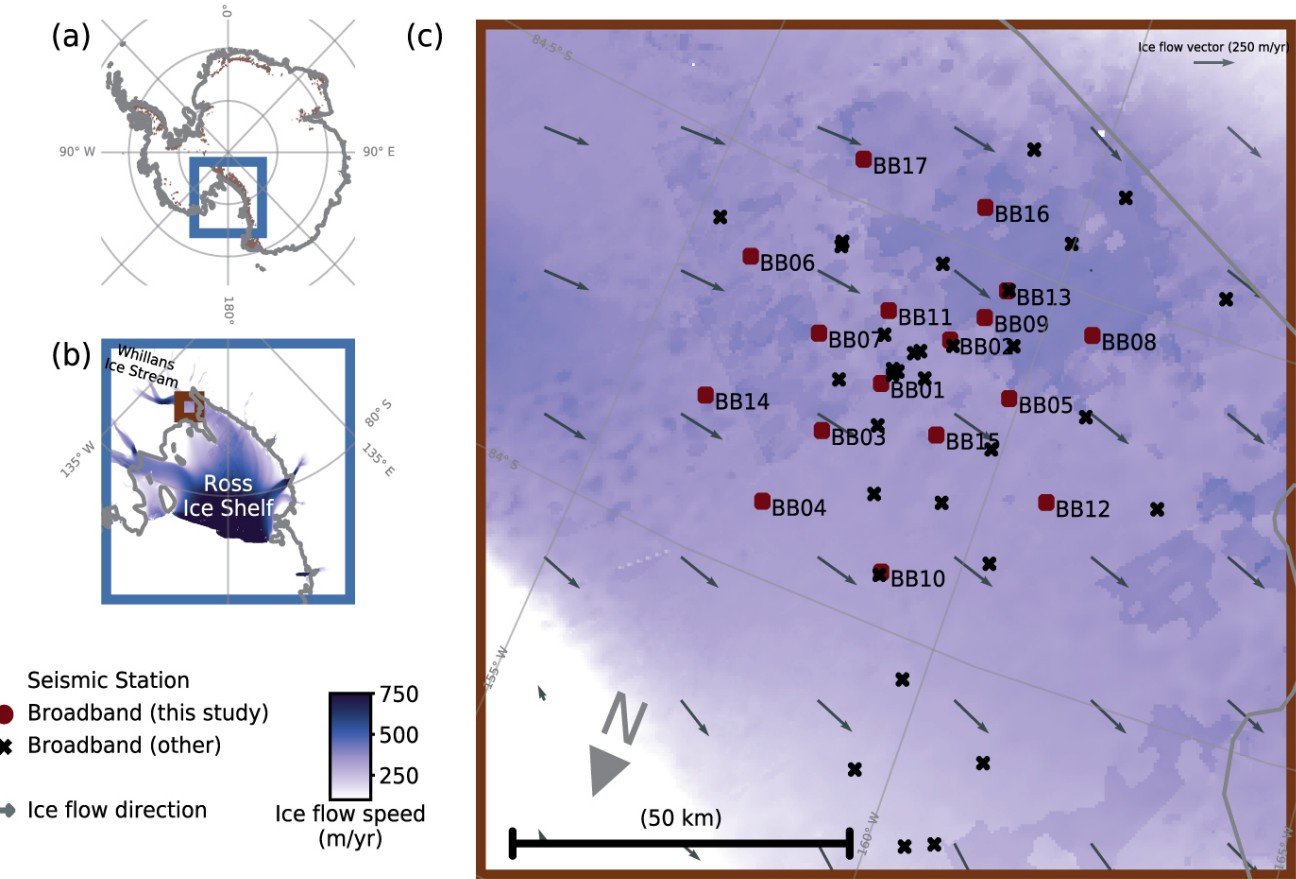

**Figure 4.** Location of the Whillans Ice Stream (WIS) and the seismic stations used in this study. (a) Outline of Antarctica indicating the location of the Ross Ice Shelf and exposed bedrock (brown) of (e.g.) the Transantarctic Mountains from Burton-Johnson et al. (2016), (b) Location of the Whillans Ice Stream in the context of the Ross Ice Shelf (BB01: 84°17'43.8066"S 158°9'47.1636"W), with ice flow speed (m/yr), (c) The WIS temporary broadband stations deployed in austral summer 2010 and 2011. Shown are the stations used in this analysis (red circles, labelled by station name) and the other deployed stations not used in this study (black crosses). Ice flow speed data are from Rignot et al. (2011). Map generated by *agrid* Python module (Stål and Reading, 2020).

## 3.2 Compiling the master event catalogue for the WIS

We compile a cryoseseismic event catalogue for the WIS using the multi-STA/LTA algorithm with optimized parameters as outlined above (Sect. 2.2). We use the waveform handling pipeline for automated analysis developed by Turner et al. (2021), adding the multi-STA/LTA as a new option, with a view to subsequent signal reconnaissance using unsupervised learning (Part B, Latto et al., 2024). We apply a trigger value of 3 and a detrigger value of 1, chosen as a set of triggers that effectively produce a catalogue that includes a sensible number of events. Two event catalogues are generated: one that lists reference event information and one that lists trace (i.e. station) specific metadata, named the *reference event catalogue* and *trace (metadata) catalogue*, respectively. The reference event catalogue is based on a reference arrival time, i.e. the first instance when at least

three seismometers (equivalent to the coincident trigger threshold) have simultaneous detections of an event, rather than the strict event origin time (as is the case for earthquake catalogues). The reference time precedes this arrival time by half the network time for the $N$ closest seismometers, where N is the coincident trigger threshold and the network time is the travel time for a seismic wave between the most distant seismometers in that group. We also take into account that multi-STA/LTA will decompose events into smaller events based on amplitude variations; therefore, it is more accurate to combine overlapping events into one event for the purpose of compiling the two catalogues. Overlapping events are reconciled in the catalogues by finding and merging times accordingly for triggered stations (i.e. start to stop times overlap plus or minus 30 seconds). Further considerations such as maintaining the correct duration for the actual event are detailed in the software (Turner et al., 2021).

The two catalogues also respectively report amplitude and energy as metrics for quantitatively describing events. The amplitude is the maximum (peak) amplitude of a recorded event and the energy is the integral of the amplitude squared with respect to time. Both metrics are in practice approximations because of the effect of the seismometer instrument-response function. In the reference catalogue, these values are determined by taking the average of the peak amplitude of the three seismometers (the exact number is a chosen variable) with the largest amplitudes. The trace catalogue lists these values by station for which the event has been triggered. Reference event and trace (metadata) catalogues for the WIS region during the 2010–2011 austral summer include stations triggered per event, network time, duration, amplitude, and energy ( Electronic Supplement).

### 3.2.1 Assigning confidence and known seismicity labels to events

Events are first assigned high or low confidence labels based on the number of adjacent detecting stations (Electronic Supplement provides documentation and corresponding assignments for the reference event catalogue). We find that approximately 35% of events are of high confidence (Table S2). We choose to include all events (i.e. high and low confidence) in this analysis because the low confidence event trends are generally consistent with those of high confidence events (Sect. 3.3 and Sect. 3.4). In this light, the concept of false detections entering the catalogue becomes less of a concern, although in some studies, analysts may prefer to work with only high confidence events. When possible, we label each event with a verified source. We use the Pratt et al. (2014) catalogue of stick-slip events for cross-comparison with the reference event catalogue. Taking uncertainties in start time into account, we label 140 events as stick-slip. Four of those events are determined as additional to the Pratt et al. (2014) catalogue from a manual reconnaissance. For our purposes, by 'stick-slip event', we refer to any segment of stick-slip rupture, where the typical WIS stick-slip episode is two to three ruptures over 20 to 30 minutes.

Using the global seismic catalogue (U.S. Geological Survey, 2022) during the 2010–2011 austral summer we find 68 events as potential teleseisms. We then label events according to how they compare to a local minimum in peak amplitude distribution in terms of arbitrary units (a.u.) at 3.5 (log, a.u.): Teleseism I ($> 3.5$), and Teleseism II ($\leq 3.5$). We thus find 32 Teleseism I events and 36 Teleseism II events.

Each event with a known seismicity label was subject to a visual review (Stick-slip and teleseismic events are available as .pdf products organized by assigned labels in the Electronic Supplement, with documentation for the plotting routine 'MyAnalystPlots'). Further information on how the labels for stick-slips and teleseisms were verified by analyst are included as well within a subfolder containing the labelled catalogues and related README text file. Catalogue users may find that an event

is high confidence but not immediately evident in a visual inspection of the time series. Multiple, successive station triggers add confidence to such events and usually show a change in frequency content on more detailed analysis. Other events that are attributable to known seismic sources but have a low confidence label reinforce that detections may correspond to changes in frequency content not evident in the waveforms. An example of an inspection plot is provided together with examples of the varied event types captured by the multi-STA/LTA algorithm (Fig. S2a, b).

### 3.3 Comparison of algorithms for real data

We compare the multi-STA/LTA algorithm, implemented with the recommended sta, lta pairs, with two runs of the recursive STA/LTA algorithm for real data from the WIS. From this comparison, we aim to determine if only one sta, lta pair is contributing to the event catalogue. If so, it would signify that running multiple sta, lta pairs does not enhance the resulting event catalogue and a recursive approach would be sufficient.

Given the recommended parameters, we use the minimum and maximum sta, lta pairs from the multi-STA/LTA set as inputs to the recursive algorithm (Fig. 5, caption). We choose these two corner cases from the multi-STA/LTA parameters for sta, lta pairs as a sensible point of comparison between the two algorithms that can be easily illustrated. The minimum recursive sta, lta pair is referred to as RECmin, and the maximum as RECmax.

In order to compare algorithms, we identify three detected events that exhibit diverse characteristics (i.e. impulsive or emergent behavior, envelope descriptors, and durations), with the time frame of each event being set by the detected multi-STA/LTA event duration. We then highlight the events detected by the multi-STA/LTA, RECmin, and RECmax approaches within the indicated time frames (Fig. 5). The frequency spectrum, and other available information, are also reviewed to aid the analysis of the event (Supplementary Materials, Fig. S3, Fig. S4, and Fig. S5).

For the event shown in (a) that begins at 2010-12-26T18:07:00 UTC (Fig. 5, left), we identify the source mechanism as basal stick-slip based on a previous study (Event #20; Pratt et al., 2014). The event (b) that begins at 2011-01-05T06:56:24 UTC (middle) has the initial impulsive arrival structure and high-frequency spectral content typically associated with a tectonic teleseismic event. The source of this event is a magnitude 6.1 earthquake occurring at a depth of 123.2 km from the Southeast of Loyalty Islands, New Caledonia (Supplementary Materials). It is possible that event source mechanisms of similar appearance are due to crevasse formation or propagation in the subsurface, or quakes in the firn layer, which generate seismic waveforms similar to those of tectonic earthquakes at teleseismic distances (Lough et al., 2015). Based on peak amplitude, this event is classified as a Teleseism I ($> 3.5$, log a.u.). The event (c) that begins at 2011-01-1109:27:07 UTC (right) corresponds to a tectonic teleseismic event, with a peak amplitude that corresponds to the range allotted to Teleseism II ($\leq 3.5$, log a.u.). The source of this event is a magnitude 5.5 earthquake that ruptured 8013 km from the WIS at 2011-01-11T09:16:09 UTC at a depth of 221.8 km.

RECmax is effective at detecting the initial arrivals of events and can capture the full length of long events (Fig. 5). In contrast, RECmin generally splits up events that are known to be continuous into separate segments. As a synthesis of the advantages of both RECmin and RECmax, the multi-STA/LTA algorithm with the recommended parameters detects the full lengths of a variety of event types.

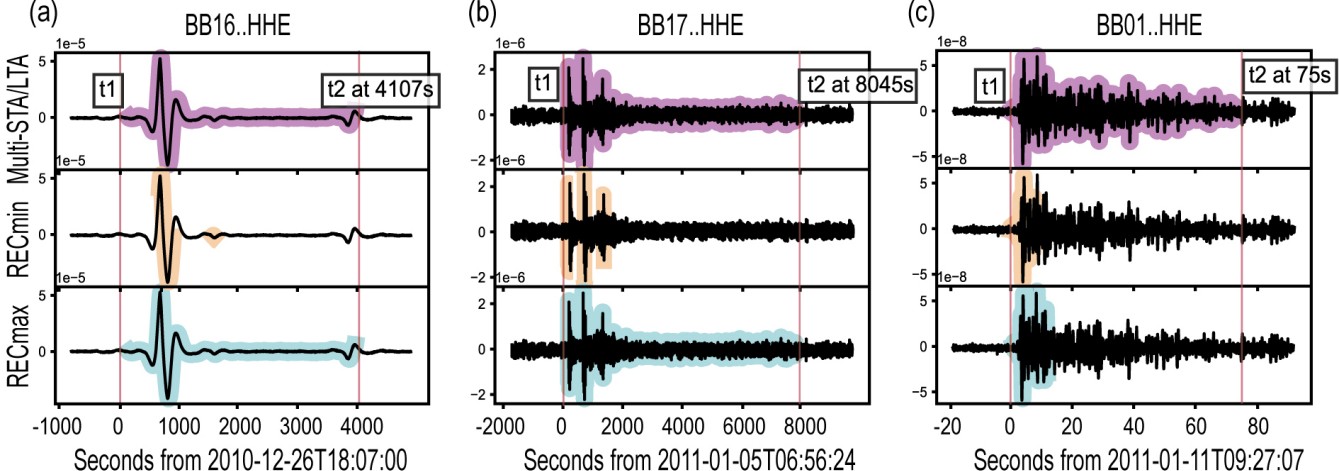

**Figure 5.** Automatically detected events shown as an overlay on manually identified events (black) to compare the performance of alternative algorithms. The y-axis shows the East component amplitude centered around zero and the x-axis shows seconds from event arrival time. The first event (a) has been identified as a basal stick-slip event by Pratt et al. (2014). The second (b) has characteristics that suggest a tectonic teleseism (Teleseism I, main text); however it is possible that other modes of brittle deformation are contributing to the signal (investigated further in Fig. S4). The third (c) also has characteristics that suggest a tectonic teleseism (Teleseism II, main text). Top row (purple overlay): detection by multi-STA/LTA, with recommended parameters, (i.e. sta = 0.03, lta = 100, $\Delta_{sta}$ = 18, $\Delta_{lta}$ = 56, and $\epsilon$ = 10). Second row (light orange overlay): detection by the recursive algorithm with RECmin parameters (i.e. the minimum sta, lta pairs sampled by the multi-STA/LTA with recommended parameters: sta = 0.03 and lta = 100). Bottom row (light blue overlay): detection by the recursive algorithm with RECmax parameters (i.e. the maximum sta, lta pairs sampled by the multi-STA/LTA with recommended parameters: sta = 0.54 and lta = 5600). The multi-STA/LTA algorithm combines advantages of the other algorithms, as it is able to capture the full length of extended events in common with RECmax.

The populations of events that are yielded by the different algorithms are compared using the reference event catalogue features: duration, total energy, and peak amplitude (Fig. 6). We use the term 'occurrence' to avoid possible confusion between count frequency and waveform frequency. Event durations from the multi-STA/LTA catalogue show a near-symmetrical distribution in semi-log space, with an equivalent number of very short and very long events and a maximum occurrence at about 10 seconds duration. RECmin, in contrast, cannot detect events greater than 300 seconds and RECmax skews towards longer events, likely missing the large subset of shorter, potentially real events detected by multi-STA/LTA. The total energy of events detected by both multi-STA/LTA and RECmin plots as a near-symmetric peak, but RECmax does not detect the lower energy events. All three algorithms show detections for approximately 50 events at very large energies, greater than 10 log (a.u.).

The distributions of the high and low confidence multi-STA/LTA events are similar for higher durations (> 1 log, seconds), energies (> 5 log, a.u.), and amplitudes (> 3 log, a.u.). The distribution of low confidence events where there are no high confidence events highlights, logically, that we have greater uncertainty in the shorter and weaker detected events. However, the low confidence event distributions also follow similar general trends as the high confidence events, lending credibility to the potentially real source mechanisms for some events that are labelled low confidence.

The distribution of the peak amplitudes provide source mechanism information that would commonly be extracted from
the magnitude of a tectonic event (such as the Gutenberg-Richter b-value; Weiss, 1997; Helmstetter et al., 2015). However, in
cryoseismology, the actual magnitude cannot be determined because the material strength, slip distance, and area of slipped
fault are usually less discernible than for crustal earthquakes. The available data and/or the seismic event may in fact be caused
by mechanisms with no slip such as, e.g., the release of dammed melt water. The maximum of the occurrences (i.e. distribution
frequencies) for all three algorithms at around 3 log a.u. can be interpreted as an upper bound for peak amplitudes resulting
from background noise (Rydelek and Sacks, 1989). After that threshold, RECmax does not tend to find larger amplitude
events. In contrast, RECmin and multi-STA/LTA preferentially detect those events. The variegated structure of the curves for
both algorithms indicate composition by mixed populations of event types.

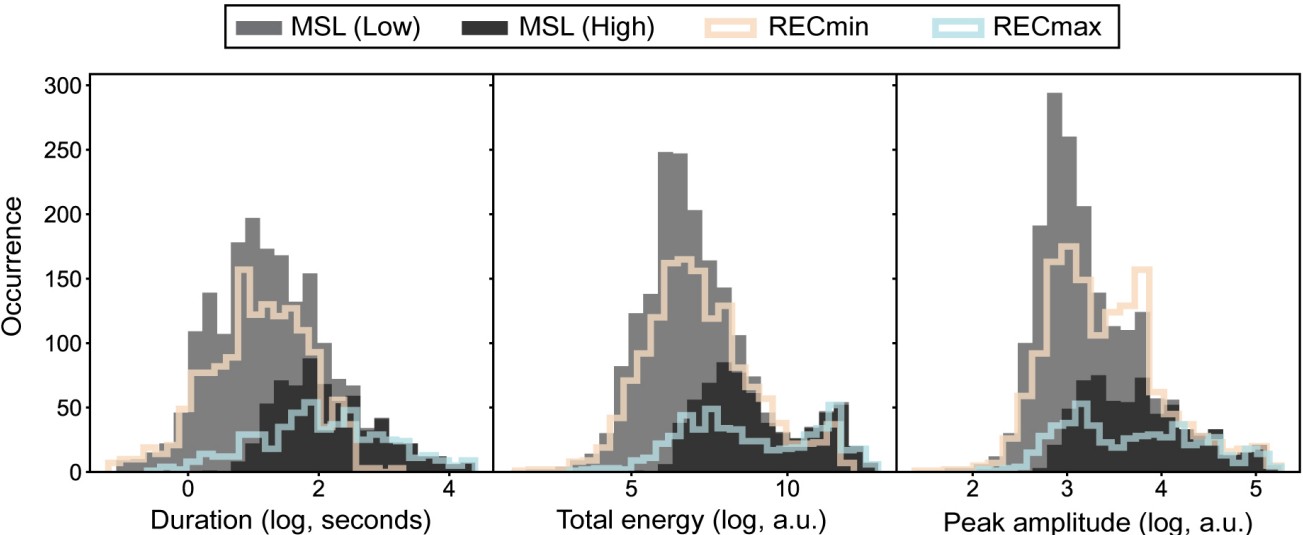

**Figure 6.** Reference event catalogue feature distributions. Shown on the x-axes are duration ($\log_{10}$ seconds), total energy ($\log_{10}$ a.u.), and peak amplitude ($\log_{10}$ a.u.), while the y-axis shows the occurrences of events with the respective feature value. Detections from the three algorithms are: MSL, i.e. multi-STA/LTA (low confidence, light grey filled, stacked; high confidence, dark grey filled), the recursive RECmin (light orange outline), and the recursive RECmax (light blue outline). The multi-STA/LTA algorithm combines advantages of the other algorithms, as it is able to match, and improve upon, the detections achieved by RECmin, while capturing the full length of extended events in common with RECmax (see also Fig. 5, parameter values given in caption). False detections are less of a concern using the current workflow than in conventional catalogue generation as events are appraised using the Part B unsupervised learning with both events and event-like noise potentially providing insight into glacier processes. See also the discussion of low confidence events following patterns of high confidence events in main text.

## 4   Discussion

The purpose of the event detection algorithm development and workflow is to facilitate a consistent approach to the generation
and analysis of event catalogues for cryoseismology and similar research.

## 4.1 Limitations

The varied nature of cryoseismicity raises the question of how an 'event' should be defined for inclusion in the catalogue. Two definitions were tested, with regard to how an event is detected by an array. In the first definition, we included a trigger condition that defined a detection when three or more seismometers triggered within 60 seconds of the same arrival time. This biased the catalogue towards events with impulsive initial arrivals, resulting in events that we know to have a single source mechanism to be truncated prematurely or divided up into separate events. As stick-slip events are known to last between 20–30 minutes, we tested (and make subsequent use of) a second definition that merges overlapping events from three or more seismometers together regardless of initial start time (Fig. S6). In all cases, the start times in the resulting catalogue indicate the time window within which the signal is observed, not the origin time of the event. No event locations are calculated or recorded in the catalogue, which is intended as a generalized reconnaissance tool. As a drawback to this approach, a small number of event groups might be catalogued under a single energetic reference event even though the source mechanisms could be different.

It is possible that events in other locations of interest for cryoseismology have event types with substantially different seismic signatures than those of the WIS (on which our simulated waveform population was based). As an example, stick-slip events can have a wide range of lengths and frequencies with documented durations from 0.1 to 1000 s and frequencies from 0.01 to 1000 Hz (see Podolskiy and Walter, 2016b). Therefore, the parameters for the multi-STA/LTA algorithm recommended in Sect. 5 can be adjusted based on individual scenarios. We recommend a fresh grid search of parameters be considered, using the same approach to the parameter search described in Sect. 5 (example codes in Electronic Supplement), prior to implementation of the multi-STA/LTA in a new location. In contrast to a trial and error approach, this method ensures a robust final event catalogue while minimizing the time typically required to find manually an optimal value for each parameter.

Use of the Euclidean norm, computed from the three-component seismic signal, has the advantage of enabling event detections irrespective of the component on which the signal arrives with highest amplitude. However, insights yielded by separate P and S wave signals could be lost in this process and we encourage analysis, following the reconnaissance enabled by the Part A –Part B workflow, that makes use of all three components. Examples include array methods and beamforming; of relevance to both impulsive events and event-like noise bursts (Gal and Reading, 2019; Gal et al., 2016; Hudson et al., 2023). Such array approaches have excellent potential for glacier studies as seismic sensor, low-power instrumentation and battery technology continue to evolve.

In this study we use the Monte Carlo approach to optimise the five key model parameters that have the strongest conditional interplay when applying the multi-STA/LTA method (sta, lta, $\Delta_{\mathrm{sta}}$ and $\Delta_{\mathrm{lta}}$, and $\epsilon$) as previously described (Fig. 2). Secondary parameters, which will vary based on study environment (i.e. background noise and seismic signal amplitudes) include the trigger and detrigger values. These values were set in this study following a brief, visual-based analysis as this was a straightforward process. Whilst any parameter choices could be optimized through the Monte Carlo analysis, the needed visualization and appraisal process for the trigger values could become unwieldy. In general, the parameters that are used should be recorded and supplied with the resulting catalogue.

## 4.2 Event catalogue analysis

We now investigate bivariate relationships in the event catalogue produced using the multi-STA/LTA algorithm (all events, Fig. 7a; high confidence events only, 7b), expanding on the univariate investigation (Sect. 3.3), and using the same event catalogue features: duration, total energy, and peak amplitude. This facilitates the exploration of event types, and thus the possible source mechanisms, that make up the catalogue. Here, the qualitative assessments of two-dimensional event types are provided to be descriptive; we further investigate event types in the companion paper to this work (Latto et al., 2024).

We choose the local minimum in the amplitude–occurrence distribution (Fig. 6) as a rudimentary threshold for a split between two large groups of events: events with amplitudes $\leq 3.5$ log a.u and events with amplitudes $> 3.5$ log a.u. As described in Sect. 3.2.1, to support catalogue reconnaissance, we have manually identified events of stick-slip origin and teleseismic events (divided as Teleseism I with amplitudes $> 3.5$ log a.u. and Teleseism II with amplitudes $\leq 3.5$), confirmed using several verification methods elaborated in the Electronic Supplement documentation. In column (b), we show that the bivariate relationships of the high confidence events follow similar patterns as the panels showing all events, and that the low confidence events occupy the shorter and weaker event domain.

The general trend between peak amplitude and duration (7, top) and energy and duration (7, bottom) of events is consistent with the positive linear association expected from cryogenic sources (i.e. increasing duration with amplitude; Podolskiy and Walter, 2016a, Figure 14). The stick-slip events present as expected (i.e. high amplitudes and energies, long durations), although there is a wide range of stick-slip durations, from 100–10000 seconds. In the bivariate relationship between peak amplitude and energy (middle), we see that the stick-slip events tend to cluster into two families of stick-slip events: one with a linear dependence between amplitude and energy and one with more variance, where events that have a broad range of amplitudes have similar energies. Stick-slip waveforms on the WIS can vary based on rupture location and tide (Pratt et al., 2014, Figure 5). Therefore, the different types of stick-slip events can be attributed to those differences. However, further analysis is necessary to investigate if the parameters of the detection algorithm perform better for particular types of stick-slip event.

The Teleseism I events show similar distributions on the bivariate plots as the stick-slip events, with a wider range of durations (0.1–10000 seconds). This spread emphasizes the diversity of earthquake sources during the WIS deployment, in terms of distance from WIS, magnitude, and depth. The Teleseism II events follow the Teleseism I events trends but into the domain of shorter and weaker signals. In the figures showing only high confidence events, the Teleseism II events cluster in a small range of durations (10–100 seconds), amplitudes (3–3.5, log a.u.), and energies (6–8, log a.u.).

The events of higher amplitude ($> 3.5$) that are detected with long durations and high energies similar to the stick-slip events could inform studies of the WIS, as they likely pertain to local, active glacier processes. We infer that these events represent more than one glacier process because we can identify several clusters of events in these bivariate spaces. For example, there is a cluster of events with very high energies and long durations (bottom). As another example in these two-dimensions, there is a cluster of events with high energies and durations between 10–100 seconds.

The events of lower energies, clustered above but adjacent to the 3.5 threshold, that occur for long durations (bottom) suggest the presence of harmonic tremors in the catalogue. Harmonic tremors are observed from ice shelf processes (such as

iceberg dynamics or ocean waves incident around ice shelves; MacAyeal et al., 2008; Cathles IV et al., 2009, respectively) or subglacial water flow beneath an ice stream (Winberry et al., 2009). Other events (lower amplitudes and energies, shorter durations), likely pertain to the noise events (Sect. 3.3). The probable source mechanisms of these event types remain to be investigated in Latto et al. (2024).

### 4.2.1 Possible tidal control

In view of the possible tidal control on the events of the WIS, we undertake a newly enabled overview analysis of this relationship, based on the 'catch-all' identification of events in the catalogue (produced using the multi-STA/LTA algorithm). We acknowledge that the length of the deployment in this study is relatively short for such an analysis, but the comparison is indicative of what carried out with records covering longer time spans. The tidal heights that we use for comparison are estimated using the Circum-Antarctic Tidal Simulation (CATS; Padman et al., 2002) and we examine the two large event groups defined by peak amplitude (i.e. $> 3.5$ and $\leq 3.5$) and illustrate their occurrence on falling and rising tides (Fig. 8). In general, similar patterns in events of high and low confidence are found for peak amplitudes $> 3.5$. The distributions for peak amplitudes $\leq$ 3.5 appear similar, but the smaller number of high confidence to low confidence events emphasizes the larger uncertainty in the analysis of weaker events.

The separation of events by falling and rising tides demonstrates that seismicity patterns are moderately correlated to the periodic tidal cycles, mostly with little difference in tide direction (increasing or decreasing), shown by the similarities in Figs. 8(a)–(c). The events with amplitudes $> 3.5$ show a tendency towards positive tide heights. The high tide tendency is corroborated in the case of stick-slip seismicity, which occurs preferentially at maximum power during high tide near the center of the stick-slip region and of the array (at 84.4°S, 157°W; Pratt et al., 2014; Barcheck et al., 2018). Conversely, events with amplitudes $\leq 3.5$ show a more symmetrical distribution. This result compares well with the observed association between relatively weak amplitude ocean microseisms and diurnal tidal cycles (Makinson et al., 2012; Anthony et al., 2015).

We further examine the context of the events by evaluating the relationships between tides and ice temperature variations with timing of event occurrence (Fig. 9). The ice temperature, retrieved for each of the 14 BBXX station positions over the deployment period, is a surface product sourced from the AVHRR Polar Pathfinder Cryosphere (Fig. 9 caption). The neap and spring tide cycles correlate respectively, to some extent, with a lesser and greater number of events per day. A possible causative mechanisms could be processes occurring within the Ross Ice Shelf in response to ocean gravity waves (Chen et al., 2019). In comparison, the ice temperature variations appear to be more weakly correlated to event occurrence. Even so, cooling through the months of available data (Comiso, 2000) may be a gradual influence on the seismic response of the WIS and the surrounding region (e.g. expected seismic response from the thermal contraction of ice; Olinger et al., 2019). The temporal patterns of daily event occurrence are similar for both the high and low confidence events. For example, the days with the highest number of overall events (e.g. Days 34, 33, 7, and 16) maintain a relative increase of events when looking at high confidence events only.

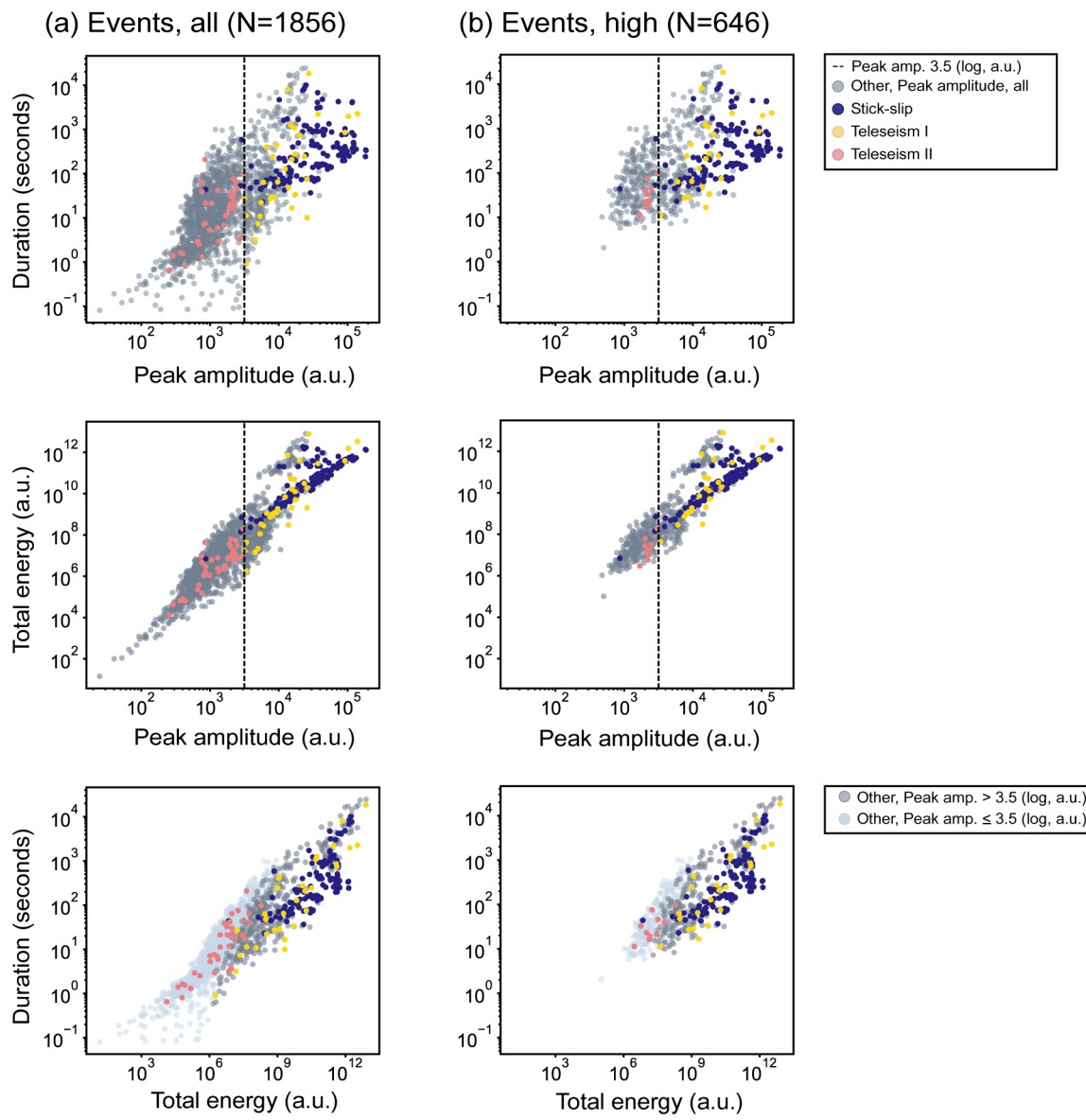

**Figure 7.** The complete reference event catalogue for multi-STA/LTA detected events shown as bivariate plots for (a) all events and (b) high confidence events only. The filled markers show all events (grey), stick-slip events (blue), Teleseism I events (yellow), and Teleseism II events (pink). In the total energy versus duration panel (bottom row), events are subdivided according to peak amplitude to show the distribution of smaller events ≤ 3.5 (light blue-grey). The threshold of 3.5 is determined from the local minimum in the amplitude–occurrence distribution in Fig. 6. Some events corresponding to amplitudes ≤ 3.5 are obscured by points corresponding to larger amplitude events and/or known seismicity labels.

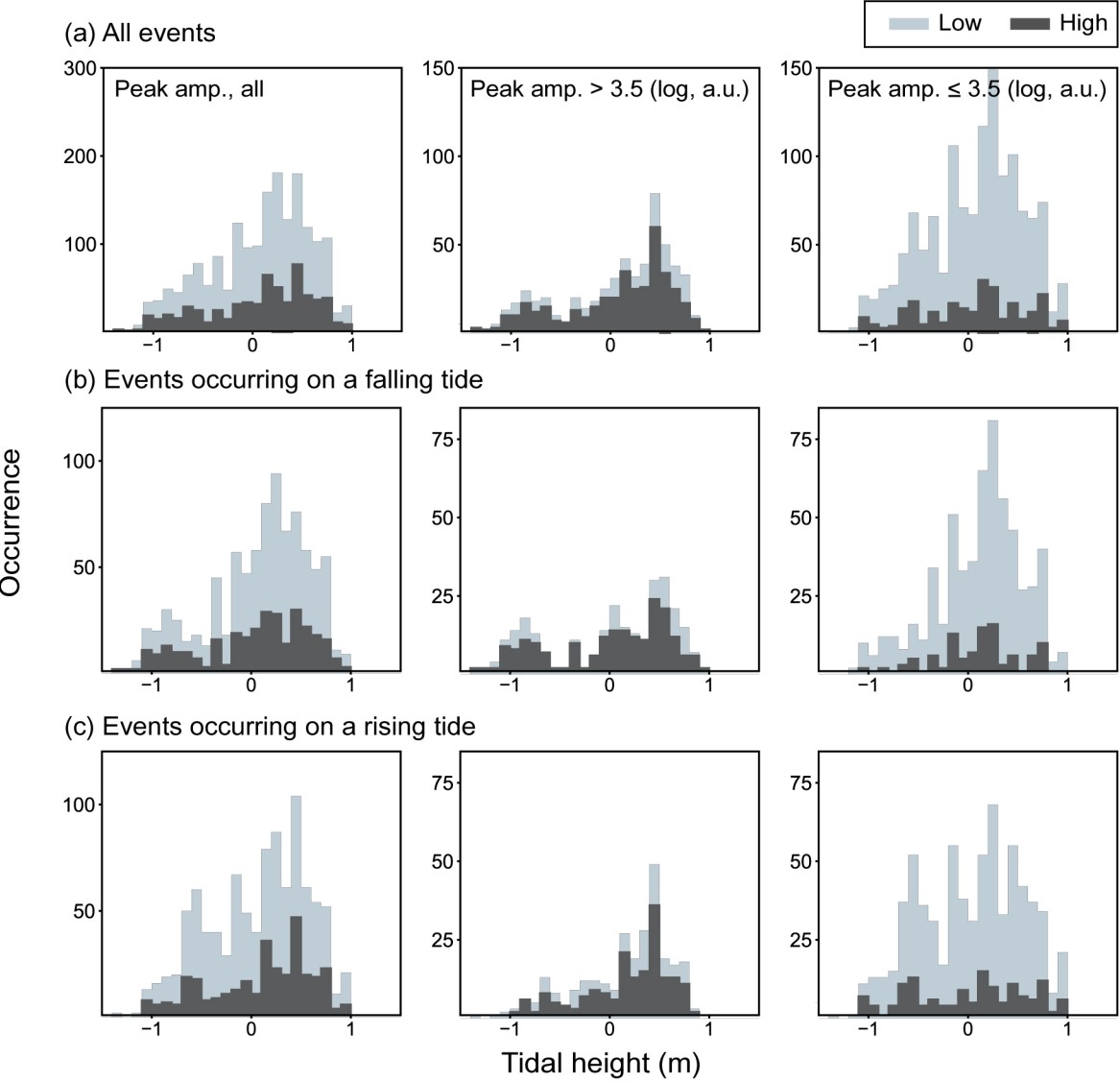

**Figure 8.** Reference event occurrence against downstream tidal height for the WIS. Events are grouped into rows for (a) All events, (b) Events occurring on a falling tide, and (c) Events occurring on a rising tide. Bars are shaded light blue-grey for low confidence events and dark grey, stacked, for high confidence events. From left to right, columns show all peak amplitudes, peak amplitudes > 3.5, and peak amplitudes ≤ 3.5. Tidal heights (m) are determined for a downstream location (84°20'20.3994"S 166°0'0"W) from the CATS tidal model (Padman et al., 2002; Howard, 2019). This location is on the Ross Ice Shelf, 59 km from station BB12 (i.e. the seismometer in the WIS array located furthest downstream). Teleseismic events are included with a view to streamlined future workflows, but are not sufficient in number to mask the temporal patterns shown (Table S2).

## 4.3 Applications

The systematic compilation of reference event and trace catalogues using the multi-STA/LTA algorithm newly enables the future application of a variety of seismic techniques, to understand glacier dynamic and hydrological processes, as the event

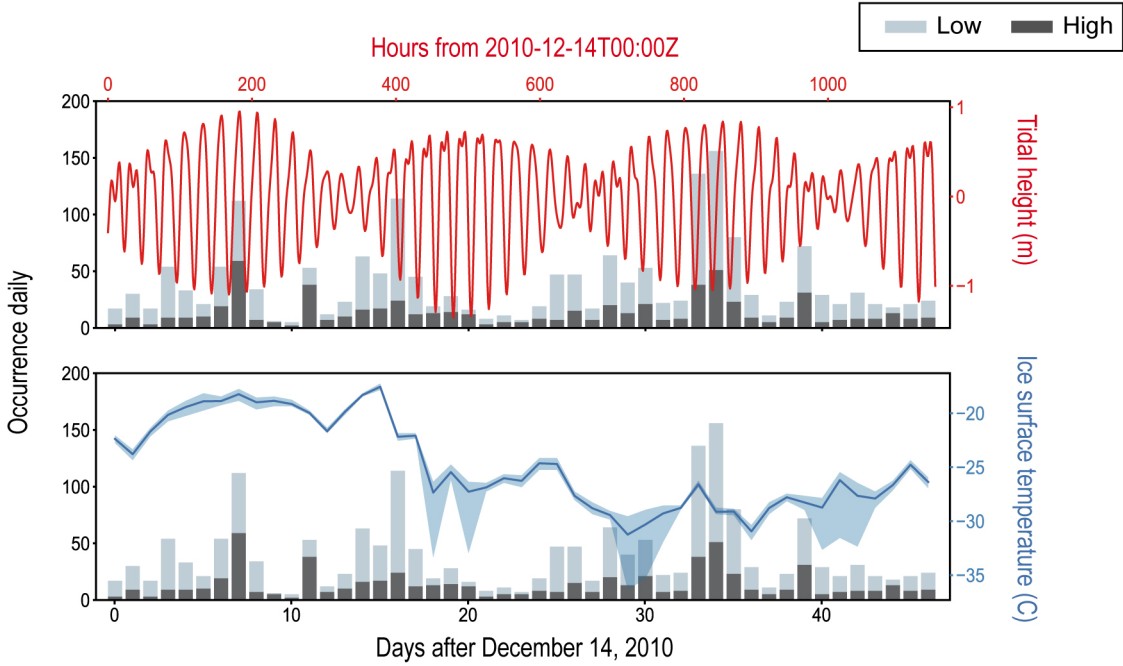

**Figure 9.** Reference event occurrence, split into daily totals are shaded light blue-grey for low confidence events and dark grey for high confidence events (stacked). Daily occurrence is plotted against (a) downstream tidal height for the WIS (red, right y-axis) and (b) ice surface temperature at all relevant stations (dark blue line is the daily average for the area and the blue shading covers the minimum and maximum station daily temperatures, right y-axis). Tidal heights (m) are determined for a downstream location, as in the caption for Fig. 8. Ice surface temperatures (C) are from the high-resolution (25 km) NOAA Climate Data Record (CDR) of AVHRR Polar Pathfinder (APP) Cryosphere, Version 2, at each of the 14 BBXX station positions (Key et al., 2016, 2019).

start time (signal arrival window) and other information are provided to the analyst. Further, the production of 'catch-all' (near-
415 comprehensive), reproducible event catalogues is a critical step towards standardized glacier monitoring as comparative studies
between locations are enabled. The algorithm and workflow may enable a more complete analysis of diverse events from longer
duration networks. In this way, new seismic deployments with 'in-ice' stations can draw on the experience gained in Greenland
and Antarctica including large-scale seismic networks like Greenland Ice Sheet Monitoring Network (GLISN) and the Polar
Earth Observing Network (POLENET) (Wilson et al., 2006; Anderson et al., 2010). The multi-STA/LTA algorithm could be
applied to these long-duration deployments to enable an expanded catalogue and optimize fill-in deployment planning. We
intend the multi-STA/LTA algorithm to be used as an additional tool in the cryoseismology toolbox, and endorse existing
approaches such as template matching or array approaches if the intent of the cryoseismology study is to examine or locate a
more specific event (glacier process) type (e.g. Nanni et al., 2022; Umlauft et al., 2023; Hudson et al., 2023).

The event catalogue produced here includes a list of the seismic stations in the array which detected each event, the net-
425 work time at detection, the duration, amplitude, and the energy. Complementing the reference catalogue, the trace (metadata)
catalogue enables manual analysis of represented stations. The new catalogue will find utility in guiding conventional glacier

seismology, taking the place of a lengthy manual reconnaissance of event types in most cases, and also pointing to any temporal patterns in event and event-like noise occurrence.

Further, investigation of event types using a machine learning approach, which is being used increasingly (Bergen et al., 2019), has been enabled, and is the subject of a companion paper (Latto et al., 2024). One of the key outcomes of our current study is that the catalogue reveals the diverse character of events from the nearby ocean and ice shelf, in addition to the events within the ice stream. Of these, many events are ambient noise, but the fluctuating noise level means that they manifest as events. Therefore, using software such as that provided in Turner et al. (2021), we aim to investigate the variety of event types using unsupervised learning based on the features computed from the seismic time series per event (equivalent to the feature sets described in Köhler et al. (2008)).

While the methods described have been developed and tested for a glacier environment, a similar workflow, including use of the multi-STA/LTA algorithm, has potential for application to other similar environments, such as volcanoes, landslides, and mining.

The semi-automated nature of the processing makes glacier monitoring using seismic methods increasingly feasible. Large outlet glaciers drain, and buttress, major ice sheets covering Greenland and Antarctica from the warming ocean. The contribution of these glaciers to sea level rise constitutes an increasing threat (DeConto and Pollard, 2016). The information from 'catch-all' (near-comprehensive) event catalogues would enable the detection and further understanding of hidden processes such as brittle cracking and basal slip, and provide improved temporal resolution of intermittent processes such as melt episodes and calving. In tandem with other information, such as that provided by satellite data, this provides a means to advance of understanding of glacier dynamics, and the response of glaciers to forcings and change.

## 5    Conclusions

We present a novel seismic event detection algorithm (multi-STA/LTA) that successfully detects events that have low signal-to-noise ratios and/or are diverse with regard to maximum amplitude and event duration. Using a Monte Carlo simulation of test waveforms and subsequent parameter search, we demonstrated how the algorithm parameters can be optimized. The algorithm's utility in glacier seismology for generating a 'catch-all' event catalogue has been illustrated through application to 14 stations from the Whillans Ice Stream 2010–2011 austral summer seismic deployment (IRISDMC; Winberry et al., 2010). The resulting event catalogues (reference catalogue, trace catalogue) encompass a near-comprehensive reconnaissance, research product that will enable further glacial seismicity studies.

We find that multi-STA/LTA is more adept than the conventional recursive approach at capturing diverse events that are characterized by a wide range of durations, amplitudes, and energies. In particular, the multi-STA/LTA approach detects events across a wide range of characteristic time scales, with durations varying by at least an order of magnitude, in contrast to implementations where the computation is based on a single set of such parameters.

The catalogue is appraised through assigning high (approximately 35%) and low confidence to events. We show that the low confidence event distributions are similar to those of the high confidence events in most cases. The significant proportion of

low confidence events for this catalogue highlights the challenges of glacial seismology in a noisy environment such as that of the Whillans Ice Stream and surrounding Ross Ice Shelf, where both local events and those external to the ice stream are potentially of interest. Many of the captured events are not immediately obvious to a visual check of the time-series, but show a shift in frequency content on closer analysis.

We demonstrate the utility of the catalogue through investigating aspects of event property distributions and links to possible signal generation mechanisms. We are able to begin analysis of the diverse event types, including stick-slip seismicity and teleseismic events, all produced from one heterogeneous catalogue. Events in the catalogue are visualized in terms of their duration, energy, and peak amplitudes. We find a partial association of seismicity with the tidal cycle, noting that a longer deployment would be preferable for such an analysis, and we consider that 11% of the catalogue are stick-slips and teleseisms (Sect. 3.2.1). We find a slight association with ice surface temperature, as an indicative example of one atmospheric observable. For both results, longer time series would be needed to support a statistical test for correlation, thereby we use the term 'association' to indicate a qualitative assessment.

The new algorithm and workflow for systematic event detection has multi-faceted potential. For conventional seismological analysis, this will enable the reproducible generation of 'catch-all' (near-comprehensive) event catalogues for cryoseismology, and facilitate further manual analysis. It will also enable progress in the wider fields of environmental and geotechnical applications of seismology. Significantly a semi-automated approach to data analysis is enabled, such that machine learning and other automated analyses may be used to enhance pattern detection and dataset exploration. Improving analysis capabilities, whether by conventional or semi-automated means, should prove to be a valuable step forward in analysing the response of remote glaciers to global change.

*Code and data availability.* The codes that produce the discussed results are written in python, are open access, and available for download from https://github.com/rossjturner/seismic_attributes (Turner et al., 2021). The codes used to produce figures and numerical results in this contribution, are available in Jupyter notebook format (.ipynb), for generalized use, in the Electronic Supplement. The multi_sta_lta GitHub repository also has stored the catalogues, the confidence assignment routine, and the MyAnalystPlots plotting routine. The Whillans Ice Stream seismic dataset is publicly available from The IRIS Data Management Center (IRISDMC) (Winberry et al., 2010).

*Author contributions.* Author 1 developed software, carried out data analysis, and wrote the text; Author 2 developed software and provided guidance; Author 3 gave overall project direction and provided guidance; Author 4 advised on the dataset and provided guidance. All authors contributed to the refinement of the text.

*Competing interests.* No competing interests are present.

*Acknowledgements.* This research was funded under Australian Research Council Discovery Project DP210100834, with additional support from DP190100418, ARC's Special Research Initiatives: Antarctic Gateway Partnership, SR140300001; and the Australian Centre for
Excellence in Antarctic Science, SR200100008. The software utilized for the analysis was based in the ObsPy python project for seismic analysis (Beyreuther et al., 2010; Megies et al., 2011; Krischer et al., 2015). We also make use of Turner et al. (2021) as the ObsPy software library that implements the multi-STA/LTA algorithm and other analytical tools, as well as event catalogue production. Figure 4 was generated by *agrid* Python module (Stål and Reading, 2020) with assistance from Tobias Stål. The Tide Model Driver (TMD) toolbox allowed for appropriate use of the Circum-Antarctic Tidal Simulation (CATS) (Padman et al., 2002; Howard, 2019). We thank colleagues Sue Cook,
Bernd Kulessa, Tobias Stål, Hannes Hollmann, Ian Kelly, and other UTAS/IMAS group members and collaborators for their contributions to discussions. We thank two anonymous examiners of the Master of Science Thesis (for Rebecca Latto, Author 1) for their careful review of the material and thoughtful suggestions for improvement.

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
