# Peer review of "Towards the systematic reconnaissance of seismic signals from glaciers and ice sheets - Part A: Event detection for cryoseismology"

_EGUsphere, 2023_

## Author Comment (AC1)

Title: Towards the systematic reconnaissance of seismic signals from glaciers and ice sheets - Part A:
Event detection for cryoseismology
Author(s): Rebecca B. Latto et al.
MS No.: egusphere-2023-1340
MS type: Research article

Please access the discussion at:
https://egusphere.copernicus.org/preprints/2023/egusphere-2023-1340/#discussion
* * *
General statement, in reply to Part A Reviewer 1:

We're very grateful for the thorough review, and we'll pay close attention to providing clarification on the detailed points that are raised. With regard to the overview intentions of our Part A and Part B contributions (respectively, catalogue generation and event reconnaissance) we see that it will be important to make small additions or rewords some parts of the abstract and introduction to make it clear that our approach has a distinct innovation, in that it takes a 'catch-all' approach to events, and noise bursts, within the seismic wavefield as a whole.  Hence, we deliberately don't pre-select the events we wish to have in the catalogue as implied by some of the R1 comments.  We instead capture the vast majority of events (broadly defined) and event-like noise bursts that occur (Part A); and then undertake a reconnaissance of these using unsupervised learning (Part B).  Given that this workflow differs to some previous literature, we see that we need to more clearly outline the thinking behind our approach, and we're of course happy to do this.

We note that Part A R2, Part B R1 and Part B R2 all returned very positive reviews, so we're encouraged that our approach is likely to be of high utility.

Below, the R1 comments are copied in grey. Author Comments continue in blue.
* * *
This manuscript present a new algorithm to detect seismic events. The method is based on the classical sta/lta detection method.  It runs the algorithm using a wide range of time windows (tSTA and tLTA) to build a hybrid detection function. This method allows to detect a wide variety of seismic signals covering a huge range of signal durations (0.1 - 10000 s).
We (the authors) confirm that these overview statements are correct.

The method is applied to a dataset from the Whillans Ice Stream. However, this manuscript does not provide new information on the source of these seismic signals.
The purpose of the m/s under discussion, Part A, is event detection specifically designed to capture seismic events and event-like noise sources in the difficult case of the glacial environment, it is not our aim in this part to investigate the source of the signals.  Part B examines the source of the signals, so we refer the reviewer to this second contribution.  We're very sorry that this key piece of information wasn't evident to the reviewer at the outset (although it is mentioned later by the reviewer in a positive light), and we we'll accordingly add an improved explanation (as per the general statement above).

Most possible "stick-slip" events are already known (Pratt et al, 2014) and their correlation with tides has already been extensively discussed.

Yes, we agree. We are presenting a semi-automated workflow across the Part A and Part B contributions, hence, we show that the new workflow concurs with previous work where the signals were manually identified and analysed.

I am rather disappointed by this paper. First, it is purely methodological with no result on glacial processes.

As noted previously, the purpose of the m/s under discussion, Part A, is event detection, it is not our aim in this part to investigate the source of the signals. Part B examines the source of the signals, so we refer the reviewer to this second contribution. See also the general statement – we're happy to outline more clearly the general thinking behind our approach.

The method could also be applied in many fields of seismology (landslides, volcanoes, ..) that also exhibit a large variety of seismic signals. It could thus better fit in a journal on seismology.

The glacier environment calls for analysis of the widest range of signal types, which is why we developed the 'Part A' algorithm for glaciology, further, the Part A followed by Part B workflow demands that the journal is focussed on glaciology. We agree that we could explain the application to a wide range of signal types more clearly, and will accordingly clarify our existing explanation.

Second, I am not fully convinced by the advantages of the method compared to other methods. The main advantage is to automatize what many researchers do by trial and errors.

The advantage of this method is that it produces a 'catch-all' event catalogue. It automates the process for many signal types and is scalable to large datasets, with repeatable results. In contrast, a trial and error approach could only be attempted for a small number of signal types over a modest dataset, further, a trial and error approach would not be so repeatable/consistent if/when carried out by different analysis over different seasons. We agree that we could emphasise the benefit of repeatability more strongly, and will accordingly add an improved explanation.

Many researchers adapt the time windows of the STA/LTA methods (Short and Long Time Average of seismic energy) in order to detect most events that can be observed by eye when looking at seismograms and spectrograms, while simultanouesly decreasing the rate of "false detections", ie, anthropogenic and environnemental noise, teleseisms or other types of events different from what they look for. But I think that this first step of looking at seismic data (on a small subset of the dataset) is essential to discover different types of events and to select signal of interest.

We agree that the data analysis should progress to extracting the signals of interest. We do this in Part B, aided by an unsupervised learning approach to expand the range of signals that can be identified. We hope that our proposed clarifications as above (general statement, and pointing the reader to Part B) will make this clearer.

Also, the method only considers two parameters (tSTA and tLTA) but does not discuss the other parameters: the minimum ratio of short and long-term energy used to define an event and the frequency band.

Our experience is that varying the parameters mentioned results in a full catalogue. We are happy to add a note in the discussion section on this point.

I guess the authors do not filter the data, while it could be an efficient way to remove noise events and to detect weaker events by selecting the frequency of interest and where the signal/noise ratio is largest.

We do not use conventional frequency-based filtering, because we are searching for events across a range of frequencies. In preliminary investigations, we also found that conventional frequency-based filtering often led to artificial event detection with our method. We do, however, pre-process the data by taking the Euclidean norm of the three-component amplitudes such that weaker signals are able to be captured by the algorithm. We are happy to add a clarification on this point.

I feel that the method allows to detect more events but most of these events are maybe noise, such increasing the work of event classification, which is the most difficult task.
We agree that we now capture many noise-like events. This is an advantage in the glacier environment, because many such signals have an ice-related source. We are happy to add a clarification on the advantage of this point. We agree that event classification is important, hence the workflow that we demonstrate in Part B.

I believe that a simpler STA/LTA method with well chosen parameter could be almost as efficient than the proposed multi-STA/LTA algorithm, while reducing the number of "false detections". At least, the authors need to demonstrate that their method detects more events but without increasing the fraction of false detections.
We take a slightly different approach, and are deliberately building a catalogue that does include a variety of signal types. The process of associating events across multiple stations ensures that only a very small number of truly spurious events enter the catalogue, although we agree that there could be a small number of co-incidental event associations. We are happy to add a clarification on this point.

The multi-STA/LTA algorithm is compared with the standard STA/LTA method, but only for the two extreme models (very short or very long time windows, l232). Why not using all models or the average model?
We compare the multi-STA/LTA algorithm with one short and one long STA/LTA model to demonstrate the benefits of the catch-all approach over individual applications of the standard approach, which are not typically averaged.

The manuscript is often hard to read and understand, many points should be clarified (see minor points below).

The manuscript stops when things could start being really interesting.
It is indeed a great shame that Part B wasn't seen by this reviewer at the outset. We hope that the concerns that arose have been addressed in response to previous comments.

What are the newly discovered "stick-slip" events? Why were they not detected before? Are they weaker than the others or do they have a different waveform? Could you try to locate these new events?
We agree that this information is hard to find. We include in the Electronic Supplement further descriptions for how the Stick-slip and Teleseism labels were verified by analyst (event_detection_for_cryoseismology/labelled_catalogues/README.txt). For the ease of access, we've copied that description below:

There are 146 events in the prototype catalogue labelled by analyst (RL) as stick-slip.

Of the 146 events, 136 fall within an assumed 30-minute duration of previously identified start times (Pratt, 2014), labelled 'STICK-SLIP, PRATT14 '. The

remaining 10 events have been hypothesized as stick-slips by an analyst during manual appraisal. These are labelled 'STICK-SLIP, PRATT14, additional.

Our assessment of whether an event looks to be stick-slip is based on the expected rupture propagation shape shown previously on the Whillans Ice Stream (Pratt, 2014). From the literature, we can expect "three separate pulses of abrupt ice velocity change during a slip event, each corresponding to the passage of a rupture front" (Pratt, 2014). These temporally-correlated ruptures are referred to as Rupture 1, Rupture 2, and Rupture 3 in descriptions below.

Of the 136 STICK-SLIP, PRATT14, 3 events that fit the requirement of occurring within Pratt, 2014 events do not correspond to visualisable ruptures. Event 20101227T172108Z potentially occurs during a third stick-slip, but it is not verifiable because of the occurrence of a coincident low-frequency event. Events 20110119T154436Z and 20101222T154843Z trigger on and off before the initial stick-slip rupture, so are likely stick-slip related but not an actual stick-slip rupture-type event. However, due to temporal correlation, we keep these 3 events in the STICK-SLIP, PRATT14  label.

For the remaining rupture-type 133 STICK-SLIP, PRATT14  events, 50 events occur during Rupture 1, 42 events during Rupture 2, 26 events clearly encompass at least Ruptures 1 and 2, and 15 events are recorded more than once in the catalogue.

Of the 10 'STICK-SLIP, PRATT14, additional' events, 6 events have uncertainty in the assigned stick-slip label upon further examination. Event IDs 20101221T040052Z, 20101221T180558Z, 20110109T145339Z, 20110122T012709Z, 20110125T004320Z, 20110126T205028Z have coinciding teleseisms and/or other low-frequency signals that confused the seismogram and spectrogram review during the manual appraisal.

For the remaining rupture-type 4 'STICK-SLIP, PRATT14, additional' events, 2 events are during Rupture 1, 1 event is during Rupture 2, and 1 event clearly encompass at least Ruptures 1 and 2. The stick-slip start times that we would contribute as additional to the Pratt, 2014 catalogue are:

Start time (i.e. ref_time in UTC)     Related event ID (Rupture #)
2010-12-15T02:02:38.887027Z     20101215T020238Z (Rupture 2)
2010-12-19T06:23:36.107027Z     20101219T062336Z (Rupture 1)
2011-01-18T21:12:15.817027Z     20110118T211215Z (Rupture 1)
2011-01-19T00:50:15.682027Z     20110119T005015Z (Rupture 1)

[Figure]

Figure1: Comparison of Whillans Ice Stream stick-slip events previously known (Pratt, 2014) and newly-detected using the multi-STA/LTA, by time of day of rupture and day of the 2010—2011 austral summer deployment. All known Pratt, 2014 events (yellow rectangles; 30-minute set lengths) are detected using the multi-STA/LTA detection algorithm, overlaid as rectangles shown from starttime to duration of event (green rectangles). Additional stick-slips are overlaid (red: verified, grey with red outline; unverified). The temporal context reveals patterns in stick-slip behavior and provides further validation of stick-slip label assignments.

What are the "tremor-like" signals mentionnent on I326?
We will refer to Part B, where the tremor events are further described, and can clarify the use of the word tremor accordingly in Part A.

Details and minor points:

Figure 1. Plots (a) and (b) could be removed, all information is also on the other subplots.
We were asked by the editor to include this figure, and we prefer to keep the subplots separate to provide a background to the STA/LTA algorithm for readers new to such detailed aspects of seismology (as usable in glacier research). We could perhaps combine (a) and (b), as a compromise that would also support a clear explanation.

Algorithm description, section 2.1.
 The classical recursive STA/LTA algorithm should be described (even it is described in the cited references) as it is the base of the multiSTA/LTA method.
We will expand the caption of Figure 1 to provide the background explanation, so this is directly available to readers, and make better use of the subplots (a) and (b) that we wish to retain.

l118. What are the values of the minimum STA/LTA thresholds (trigger and detrigger) used to define an event? How are they chosen? Why not optimizing these parameters as done for the time windows?  Fig 1 suggests the threshold is fixed at 3 and is the same to trigger an detrigger an event. Did you try other values?
We agree that the choice of trigger and detrigger value is not clearly outlined in the m/s and supplement. We apologise for that oversight, and will add a short clarification in the text. We note that the low confidence events follow the same patterns as the high confidence events, so inclusion or exclusion of the very lowest confidence events is not critical to ongoing usage of this (or any similar) catalogue.

For the reviewer's understanding: The trigger and detrigger value we use in the illustrative example in Fig. 1 differs from that chosen for the rest of the paper. In Fig.1, the trigger value is 3 and the detrigger value is 0.5, which were picked to show the best comparison between event detections for this example waveform.  In the application of multi-STA/LTA on the Whillans Ice Stream (Sect. 2–4), the trigger value is 3 and the detrigger value is 1. Referring to the application of multi-STA/LTA, we did test multiple solutions of the trigger and detrigger to ensure that the number and nature of triggers was reasonable in a visual inspection. In principle, the related variables: trigger pairing and the number of seismometers required for a valid detection, could be constrained using a Monte Carlo simulation. However, we optimise these parameters visually, instead of as done for the time windows, due to the need for a wide view across several representative days.  As above, we'll add an abridged version of this explanation.

I don't understand eq. (1) and point (3) (l116). Is the hybrid function the average (as in eq(1)) or the maximum value (l116) of all single-parameters STA/LTA functions?
Many thanks for bringing this to our attention.  We confirm that point (3) states that we take the maximum of the characteristic functions (which we do) but the equation shows a summation. We will correct this. The maximum function is used rather than a summation to avoid reinforcing detections from the time windows in the middle of are range of parameters (which will likely have detections in multiple nearby windows) in preference to those at the extreme of our range of parameters.

 l133. I don't understand what represents epsilon?
We agree that the description of epsilon requires further clarification in the text. The purpose of the epsilon parameter is to ensure computational efficiency by not calculating more windows than necessary. The parameter is a tolerance value that ensures the most spread out of the sta or lta windows is not spaced closer than a factor of epsilon (i.e. we do not go smaller than this tolerance).

Section 3.2 should be moved to the "method" section 2. It describes how the catalogue is compiled and is not specific to the Whillans Ice Stream catalogue.
We will keep Section 3.2 as is to maintain the linear workflow, but will consider the naming of the relevant subsections to improve clarity.

 l194: I don't understand this sentence: " The reference time precedes this arrival time by half the network time for the N closest seismometers."  What is the "network time"? What is the value of N?

We agree that N, and the network time are definitions generally, are not currently defined in the text and that will be added. N is the coincident trigger threshold and the 'network time' is the travel time for a seismic wave between the most distant seismometers in that group.

l195 "We also take into account that multi-STA/LTA will decompose events into smaller events based on amplitude variations." I think this is a drawback of the method; and a simpler STA/LTA method with well chosen parameters and with a detrigger threshold lower than the trigger threshold may avoid this problem. You should describe in more details how you merge "overlapping events" into one event to obtain the correct event duration.

We address the background to this concern firstly in our general statement (above), and emphasise that we turn this challenge into an advantage: we are able to explore the complex and varied glacier seismic wavefield in a systematic way and this wavefield includes signals with a lower signal to noise ratio than in conventional analysis. Secondly, we are happy to provide clarification on our carefully thought-through procedure with regard to overlapping events.

l213 "Taking uncertainties in start time into account, we label 140 events as stick-slip. Four of those events are determined as additional to the Pratt et al. (2014) catalogue from a manual reconnaissance." How do you know that these 4 events are "stick-slip"? Could you show examples of a newly detected "stick-slip" event and a known stick-slip event?

We refer R1 to the above reference of the Electronic Supplement and included description of our procedure for stick-slip manual identification.

l219 "What means "a.u.": arbitrary unit?
We are happy to add the meaning of a.u. into Sect 3

l267 "The distribution of the peak amplitude occurrences provides source mechanism information". Could you specify which "source mechanism information"? Do you mean the Gutenberg-Richer b-value?
Yes, we are referring here to the GR b-value. We will clarify the text and reference to the Weiss, 1997 citation.

l268 "However, in cryoseismology, the actual magnitude cannot be determined because the material strength, slip distance, and area of slipped fault are usually indiscernible." I don't think this is a big issue and a major source of uncertainty on magnitudes. The problem is even worse for earthquakes that occur at depth were material properties are less constrained.
We're happy to modify the sentence to '… are usually less discernible than for crustal earthquakes.'

l271 "The maximum of the occurrences ". Do you mean the maximum of the distribution of event amplitudes?
There are several similar sentences where I was not sure to understand what the authors mean.
We are happy to clarify the use of the word choice of occurrence with a reference to Fig. 5.

l287 "As a drawback to this approach, a small number of event groups might be catalogued under a single energetic reference event even though the source mechanisms could be different." Yes indeed, many processes produce a wide range of signal peak amplitudes (Gutenberg-Richter law). I think that the event duration or frequency content is more useful to identify the source process.
Our intention with this sentence is simply to acknowledge that a small number of events might coincide. As previously noted, the reconnaissance of source processes is discussed in Part B.

l289 "It is possible that events in other locations of interest for cryoseismology have event types with substantially different seismic signatures than those of the WIS (on which our simulated waveform population was based). " Yes, indeed. For instance, basal stick-slip events have duration ranging between 0.1s and 1000 s and frequencies between 0.01 Hz and 1000 Hz (see Podolskiy and Walter 2016 for a review).
Thanks for this suggestion – this is a good example to note. We will add a half-sentence accordingly.

Fig 6 "The multi-STA/LTA algorithm combines advantages of the other algorithms, as it is able to match, and improve upon, the detections achieved by RECmin". But are all detections real events or is there a significant fraction of false detections (noise)?
Our general statement (at the top of this reply) addresses this point, and we are happy to add a clarification that our definition of events is broadly defined, and includes noise-like bursts, which likely have a glacier process origin (including the adjacent ice shelf, in the case of the WIS).

Fig S3: The signal with a frequency of 0.01 Hz is consistent with stick-slip, but it could also be a teleseism
We agree that there is potential that this is true. However, we carefully confirmed the label using Pratt's catalogue and our own method for flagging potential teleseisms.

Fig S5: Why filtering the signal? At least the spectrogram should be shown for the raw unfiltered signal.
We chose to filter the signal because the low frequency rumblings were hiding any discernible signal in the illustrative figure. Adding the raw spectrogram to the panel won't be informative in this case, so we prefer to add a note to the figure caption in view of transparency.

l303 "we have manually identified events of stick-slip origin": could you explain how you distinguished "stick-slip" events from other types of events?
See the above extract Electronic Supplement and included description of our procedure for stick-slip manual identification. We're happy to add a clearer pointer to this.

l306 "The general trend between peak amplitude and duration (7, top) and energy and duration (7, bottom) of events is consistent with the positive linear association expected from cryogenic sources " This is very general and true for many different source types. So it is not useful for classifying events.
We agree that in the figure the overlapping linearity by event types is shown; however, the ability to analyse events in a multi-dimensional feature space is useful for the multi-dimensional machine learning technique applied in Part B.

l325 "The events of lower energies,..., that occur for long durations (bottom) suggest the presence of harmonic tremors in the catalogue." This could be interesting! Could you show an exemple of seismic signal and spectrogram ? What could be the source (slow-slip event, water flow, storm ...)?
We will refer to Part B, where the tremor events are further described, and can clarify the use of the word tremor accordingly in Part A.

Section 4.2.1. The correlation with tides and temperature is interesting and is a good way to investigate the source mechanisms. But it should be done after classifying events in different types, ie, removing known stick-slip and teleseisms.
We agree in general, and that it is relevant to retain stick-slips, but teleseisms are likely to occur without any association to local tides, hence, they might slightly lessen any tidal or other association, but are not likely to impact any conclusion or insight drawn on this point.

Rather than amplitude, I think that frequency content (average frequency and width of the spectrum) and signal duration could be better parameters to discriminate source mechanisms.
See companion paper, Part B, for further discussion on this point.

Fig 7. You could show only plot (a) as it contains almost all information shown by the other plots.
We prefer to retain the separate plots to allow the reader to view the content with better clarity.

l361: "Further, the production of near-comprehensive, reproducible event catalogues is a critical step towards standardized glacier monitoring as comparative studies between locations are enabled." I agree that comprehensive and reproducible catalogs are valuable, but I think that standard methods (simple STA/LTA or template matching) can already produce such catalogs.
We address this point in our general statement, which hopefully explains why our algorithm and extended catalogue is a valuable additional approach, especially in reconnaissance analysis of the complex wavefield and variety of events that it contains. We would be enthusiastic about a template matching approach for more detailed analysis of a specific event type, and are happy to add clarification to the discussion accordingly.

I think that each glacier is different and that all algorithms and parameters need to be adjusted for each case study. I also think that using different methods may allow to detect events that are still unknown. The main problem is not the detection of events, that can be easily automated and reproduced by others.
We address this point in our general statement, and hope that our workflow Part A and Part B will allow a more general approach that yields additional insight.

The classification of events is much more tricky and subjective, often done "by eye" without objective criteria. I understand that this is the goal of your companion paper (l373) and I am very interested to see how a fully automatic machine learning method can perform.
This is the first mention of the Part B paper by this reviewer, so we refer to our general statement, and hope that other readers will now understand the intentions behind our workflow at the outset.

l370 "The new catalogue will find utility in guiding conventional glacier seismology." Can you explain how?
We again refer to our general statement, and will add clarification at this point.

L408 "We find a partial association of seismicity with the tidal cycle,". This is not a surprise since the catalog contain many already identified stick-slip events that are known to be driven by tides.
In 3.2.1 we discuss that the majority of events in the catalogue are not stick-slip nor teleseisms. Of the 1856 events (broadly defined) 140 are stick-slip and 68 are teleseisms making up 11% of the catalogue.

L12, L409 " We find a slight association with ice surface temperature, as an indicative example of one atmospheric observable.". I don't see such a correlation when looking at Fig 9. This "association" should be quantified and tested using a statistical test.
We confirm that we chose our words carefully, in this instance, and are happy to add a clarification that a longer time series would be needed to support a statistical test or more robust statement.

L414 "semi-automated approach". When reaching the conclusion I still don't clearly understand which part of the detection method is not automatic?
We use the term 'semi-automated' as the (automated) 'multi-' algorithm is not applied in isolation, but is done so alongside some actions of a human analyst. This have been described in the article, and include the pre-testing of algorithm parameters. More generally, the Part A, Part B workflow is best described as

'semi-automated' because some external information is included in the analysis, such as the likely times of arrivals from teleseismic events.  The essence is that a much-extended reconnaissance of the seismic wavefield by a human is enabled by the 'catch-all' catalogue generation followed by the unsupervised learning.

---

## Author Comment (AC2)

Title: Towards the systematic reconnaissance of seismic signals from glaciers and ice sheets - Part A: Event detection for cryoseismology
Author(s): Rebecca B. Latto et al.
MS No.: egusphere-2023-1340
MS type: Research article

Please access the discussion at:
https://egusphere.copernicus.org/preprints/2023/egusphere-2023-1340/#discussion
* * *
Below, the R1 comments are copied in grey. Author Comments continue in blue.
* * *
The authors present an exciting study that describes a "recognisance" algorithm for detecting seismicity from a range of seismic signals. The method is novel and the concept of a recognisance algorithm that is sensitive to a range of seismic signals will likely be of much use to the community.
 Many thanks for this very positive appraisal – we confirm that the emphasis in this work is capturing the range of seismic signals (Part A) in a reconnaissance of the seismic events (and likely following analysis, as in Part B), and event-like noise that is present in a glacier environment. (A minor confirmation, we carry out 'reconnaissance' information capture, whereas 'recognisance' may have been inserted by a spell-checker, and is a different/legal term).

However, although I would like to see this work published, I think the wording of the manuscript needs some work before it is ready to accept for publication. Firstly, the novelty of the algorithm needs to be clarified. This is probably a minor point, but at the moment the method appears to already be published (Turner et al., 2021), but the authors do not make this explicitly clear throughout the paper. I'd like to see the original methods paper properly acknowledged where the method is first introduced, and the tone of the paper changed to reflect that the method is applied to cryoseismology here rather than introduced as a new method.
 We're happy to add a clarification as follows:  The Turner et al., 2021 citation refers to a (properly documented, code-reviewed, software library) handling framework for waveforms as a pipeline for automated analysis (i.e. machine learning).  It was developed by the same research team, and the full reference is here:
 Turner, R.J., Latto, R.B. and Reading, A.M., 2021. An ObsPy Library for Event Detection and Seismic Attribute Calculation: Preparing Waveforms for Automated Analysis. *Journal of Open Research Software*, 9(1), p.29.DOI: https://doi.org/10.5334/jors.365
 This software pipeline has multiple options for the event detection algorithm (including the standard STA/LTA). The current submission 'Part A' is, accordingly, the correct reference for the 'multi-STA/LTA' algorithm as a novel approach, and we confirm that it was designed primarily for cryoseismology (although it could well be useful for other environmental seismology studies also).  Going forward, other users could also select the 'multi-' algorithm (for example) if they use the Turner et al. 2021 software pipeline.

Secondly, I remain to be convinced by the concept that such a deliberately broad method can outperform a more specific method suited to one task (e.g. basal icequake detection). Perhaps it can, but I see no evidence in the paper to back up the claim made in the conclusions that the method presented can provide a "near-comprehensive event catalogue".

We agree that we need to define what we mean by 'near-comprehensive' (wide range of event and event-like noise types). Should a research task be focussed on a specific event type, another algorithm may well be more appropriate as this would avoid having to handle other event types in the same catalogue. The currently presented algorithm aims to capture a wide range of seismic signals, prior to semi-automated ongoing analysis. In fact, basal stick-slip events -are- very well captured by the multi-STA/LTA algorithm and we are happy to add a comment on this point.

To conclude, I think this work will be a valuable contribution to the field and I don't find any issues with the results themselves. However, the text needs to be somewhat revised to tone down the claims made. Assuming the authors are happy to do this, then I would be very happy to see this paper published. We apologise if we seemed to be overstating any point. Hopefully the amendments (as per above and comments below) provide suitable background re: the novelty and scope of the current study, and intended future contribution of the reconnaissance workflow, in the context of other options).

**Comments:**

The introduction does seem to be written in rather a bold way (e.g. "Since STA/LTA and correlation-type algorithms have enjoyed only limited success when applied to environmental seismology"). Both STA-LTA and cross-correlation methods are effective when used in certain ways, as the authors elude to and indeed that is the premise of this paper. I'd suggest toning down the limitations of these algorithms a little, since actually there has been much success in using these algorithms carefully, as part of broader methods. Indeed, I cannot think of any passive seismology method (other than manual detection) that is not at least somewhat built upon one of these two foundations. I think it would be useful to outline the scope of the study from the very beginning of the introduction, to clearly clarify the scope of the work to the reader before making the perhaps bolder claims.
We are happy to reword as suggested. This will be partly covered in response to Reviewer 1 (see our general statement in response to that reviewer).

From the introductory text, I was expecting Section 2.2 to be its own section. However, I can see the logic behind having it as a subsection of the overall algorithm method. Regardless of the structure, I think it would be useful to provide more details on the analysis of the algorithm testing results presented in Fig. 2,3. There is a lot of information held in those two figures, but they are not yet adequately described in the text. In other words, I was excited to read that the algorithm would first be tested on synthetic data, but was then left a little disappointed that the key findings are buried in the figures without much explanation of what they show.
This information is included in the journal and electronic supplements to avoid the main text. We are happy to provide better pointers to this material in Section S2 in the Supplement.

L183-187: Perhaps it is common to only use the vertical component. However, best practice for any body wave data should be to use both vertical and horizontal components. However, mixing the three components via the Eucllidean norm might cause one to loose information about whether a phase is a P or S phase. While information loss is not inherently a problem for detecting events, incorrectly identifying S-wave phase arrivals as P-waves would result in false detections, since they are likely from the same event. I'm not suggesting the authors should revisit this component of their method, but I think they should be clear about the limitations, especially the possibility of false triggering. I'd suggest it would be beneficial to also discuss somewhere how their method might be developed further to act on the vertical and horizontal phases separately. If the authors would like any pointers to literature describing how to use

the vertical and horizontal component information for P and S wave association in detail, including how a firn layer can affect such results, then this paper and references therein provide further details (https://doi.org/10.5194/egusphere-2023-657 ). (No requirement to cite this work, just a potentially useful paper that covers the point raised).

This is a good point, and we are happy to add a few sentences to the discussion section in this regard. We're fine with citing the suggested paper (at the time of writing, it is in the final stages of review), and might perhaps also add a further citation that certainly supports being able to detect/work with S waves (https://doi.org/10.1093/gji/ggw150), and also following work with the same lead author (not included in the previous suggested reference list).

L189-190: Some clarification on the novelty of the method is required. Until this point in the paper, I was under the impression that the authors were presenting a new algorithm for event detection. However, from a glance at Turner et al. (2021), it looks like the value of this work is more in showing how the cited algorithm is implemented? Could this be clarified, and if the implementation is originally from Turner et al. (2021), then that paper be clearly referred to in Section 2.

See above clarification re: the Turner et al. (2021) software library. The first impression is correct.

L200-202: Perhaps too technical a question, but more for my interest: Are the signals instrument-response corrected before being passed through the algorithm? If not, then "energy" might be better referred to as "an approximation of the energy", since different frequencies might exhibit different amplitude responses dictated by the instrument transfer function (response). Normally I wouldn't raise this point, but since the authors are attempting to describe as broad an event detection algorithm as possible, frequency response could become important in certain instances. Maybe at least worth making readers aware of this point.

We are happy to add a comment as suggested.

Figure 5: It would be nice to see more detail in each of the time-series. I cannot dicern any differences from the plots in this figure. Maybe for each, the authors could include an inset figure zooming in on any differences in the first arrival, perhaps plotting all three signals over one another? Otherwise I'm left questioning what difference the multi-STA/LTA algorithm makes. In summary, I imagine it is better, but the figure does not currently communicate that.

We intend that the reader will focus on the difference between the purple, light orange and light blue overlays, which are clearly different (assuming the .pdf is showing OK on the screen or printout). The waveforms should be the same.

L276-277: Great aim. However, this sentence is very much left hanging. I'd like to see some text introducing how the authors will justify how the work presented here meets that aim in the remaining discussion.

This should be better supported following the clarification made in response to Reviewer 1 (general statement). Also, we're happy to expand on this point in the 'Applications' section.

Section 4.1: The biggest limitation of this work to meet its aim is that such a broad search algorithm does likely not perform as well for certain event detection scenarios, compared to more specific methods. For example, a coalescence/stacking based migration algorithm will, by design, be better for filtering out false triggers caused by noise. Therefore, taking basal icequakes as an example, the authors method is unlikely to outperform a migration method. This does not detract from the method presented by the authors, since a general recognisance method is definitely very useful in many contexts. However, it is definitely a limitation that needs to be mentioned, because it really does present a barrier for "a consistent approach to the generation … of event catalogues" (L276-277).

Again, this should be better supported following clarification made in response to Reviewer 1 (general statement). They key point is whether the event detection intends to capture a wide range of event or event-like noise types (as in the current study), or whether the nature of the event in question is previously known.

L411-412: This statement is definitely too bold. I am not convinced that one can ever develop algorithms that produce "near-comprehensive" event catalogues. However, more specifically, a general recognisance algorithm, which this is presented as, is unlikely to outperform a specific algorithm for a specific purpose. The algorithm presented here is definitely a valuable contribution to the field, but I see no evidence in the manuscript that it can produce "near-comprehensive" catalogues of seismicity.
We are happy to amend the text to be consistent with previously addressed comments, as in the above point, and also following a careful definition of 'near-comprehensive' (wide range of event types), or a wording change.

**Minor comments:**

There should be spaces between units (e.g ma^-1 should be m a^-1, otherwise it is technically milli years).
Corrected accordingly.

L19: "exceptional" – rather emotive language. Consider removing.
Happy to remove this word.

L23-24: I think the definitions of the two event detection types are a little narrow. I'd view STA/LTA algorithms as just one subclass of any algorithm that searches for a peak in energy with particular frequency content (related to the length of the STA window). These can then be used to detect phase-arrivals, or be used in combination with more sophisticated phase associators, or coalescence-based algorithms to improve detection. Perhaps worth somehow very briefly mentioning this (as an STA/LTA algorithm on its own is not very useful/causes lots of false triggers).
Happy to follow this suggestion.

L23-24: Further to the point above, one could consider array-based methods as a common earthquake detection type too? Not sure if this is worth mentioning, but picking phase arrivals on individual channels is only one class of earthquake detection. If the authors do decide to mention this, then examples of some relevant papers are included in the introduction to this paper: https://doi.org/10.5194/egusphere-2023-657. (sorry to refer to this paper again – just easiest way to point to the relevant literature cited within it).
We're very familiar with array analysis, and are happy to include the suggested reference, and also other earlier relevant citations not in the suggested reference in support of this point more generally, e.g. https://doi.org/10.1029/2018JB015526 and book chapter 'Beamforming and polarization analysis', Gal and Reading (2021) in Seismic Ambient Noise (Eds Nakata, Gualtieri and Fichtner).

L43-44: I'd say that cross-correlation algorithms are inherently also similarly prone to missed detections since they are typically based on the assumption that similar events occur within a catalogue.
Happy to amend wording accordingly.

L72: Is QuakeMigrate a spectral-based method?
Apologies, this occurred inadvertently, when making some of the background text more succinct.

QuakeMigrate uses migration based techniques that use coherency and waveform stacking in detection. We'll fix the wording.

L122: "event" – would it be perhaps better to refer to it as an "event phase arrival"?
Happy to amend wording accordingly.

Figure 1: Isnt the data in Figure 1b also plotted on Figure 1c? Not sure this section is therefore required. Similarly, Figure 1a could also be removed as all the information is contained in (d).
We were asked to include this figure by the Editor, as a step by step illustration of the STA/LTA concept. Hence, we prefer to keep the separate figures. We're happy to follow further editor guidance, perhaps combining panels a and b as a compromise.

L170: The minimum distance rather than the maximum distance is probably the more relevant number.
Happy to amend word choice from maximum to "up to 600 km" (L170-171) to follow the exact statement in the quoted reference (Wiens et al., 2016).

L199: That reference is not to software documentation, but to a paper describing the software. Pointing to any software should really be done via a software repository DOI in the Acknowledgements. However, I think the authors are actually refering to the paper here, and so should simply remove the word "documentation".
Happy to amend wording accordingly.

L236-237: I think this sentence doesn't make sense with the word "event" at the end. Apologies if I misread, but would it be possible to reword if indeed it doesn't make sense?
Happy to amend wording accordingly.

Figure 5: Add subplot labels, then refer to accordingly in the text (rather than left, right etc). That way the UTC time stamps can be remove from the text. Also, the caption is too long. Consider shortening.
We are happy to follow this suggestion.

---

## Author Response (AR2)

Title: Towards the systematic reconnaissance of seismic signals from glaciers and ice sheets - Part A: Event detection for cryoseismology
Author(s): Rebecca B. Latto et al.
MS No.: egusphere-2023-1340
MS type: Research article

Below the Editor and Reviewer 1 Comments (Second Review) are copied in grey with Author responses continuing in blue.
* * *
Editor Comments

Thank you again for submitting your revised manuscript entitled "Towards the systematic reconnaissance of seismic signals from glaciers and ice sheets - Part A: Event detection for cryoseismology".

Your manuscript has now been seen by 2 referees. While Referee #2 was fully satisfied by your revisions and replies to the other referee and recommended this work for publication as is, some important points were raised by Referee #1 (please see below). In brief, Referee #1 did not find your previous replies satisfactory, questioned the innovative nature of the paper, and was not convinced by the demonstration of the advantage of your approach versus other algorithms. The Referee also raises additional specific points that need to be addressed for the paper to be acceptable for publication.

**Response to Editor**
Many thanks for this reply. We are cautiously enthusiastic about the high potential of the workflow that we present, for the reconnaissance of mixed glacier/ice sheet environments, across the Part A and Part B papers. We are also pleased that the two reviewers for Part B now recommend publication. We're very pleased that Referee #2 was fully satisfied by our revisions to our Part A manuscript.

We're happy to address Referee #1's second review and have structured this according to the Editor's summary, that is:
a) innovative nature,
b) advantage of approach vs other algorithms (includes response to previous replies) and
c) other points raised.

We hope that following the changes outlined below, especially noting our 'Prelude' clarification of the difference between reconnaissance and high-resolution studies, our Part A m/s will be considered suitable for publication in TC.

(... continued overleaf)

**Response to Reviewer #1 (second review) Comments (reordered as per the Editor's summary)**

**Prelude)**
As a key clarification for this second revision (in response to Reviewer 1), we newly expand the distinction between the 'reconnaissance' scope of our method for large glaciers and widely spaced recording sites, and the 'high-resolution' capability of other work (we add the suggested 'Nanni et al., 2022' and related references). We also add text regarding the utility of our Part A – Part B workflow for targeting subsequent deployments, suggesting that readers could logically make use of such high-resolution methods (as in the references suggested by Reviewer 1) in locations targeted using our workflow. Clarifications in the m/s are noted as part of the response below.

**a) Innovative nature**

o    … just I don't think this study is very interesting and innovative.
o    The method is not new, it has already been published (Turner et al 2021). This work is only an application to a special case study.

**R1.2ndRev.01**
Our focus is on expansive glaciated areas, and/or widely space recording sites, and is innovative (together with the Part B machine learning component) in progressing knowledge for those settings. To correct a misunderstanding, we confirm that the current m/s *is* the first presentation of the multi- STA/LTA algorithm, i.e. it has not been previously published. This is explained in the main text, lines 201-203, quoted below, so we make no change but we are happy to rephrase (or clarify at a different point in the m/s) if there is some ambiguity remaining.
(Extract from line 201) We use the waveform handling pipeline for automated analysis developed by Turner et al. (2021), adding the multi-STA/LTA as a new option, with a view to subsequent signal reconnaissance using unsupervised learning (Part B, Latto et al., 2023).

**b) Advantage of approach in comparison to other algorithms**

o    What is an "event" anyway?

**R1.2ndRev.02-1**
We agree that it is important to be clear what is meant by an event, and therefore move the definition (previously provided at line 98) to the earliest logical point in the main text. We also re-order some text from the paragraph beginning at line 26.
Text inserted at line 27 (after '… noise'): We use the term 'event' broadly to include impulsive signals, and waveform changes (such as an amplitude increase or frequency content change) with a less distinct onset. In some glacier environments, event-like noise is of as much interest as impulsive cryoseismic events, as both signal types yield insight into glacier and/or ice shelf processes.
Also, we modify the text previously at line 98 to avoid direct repetition. Modified text:  Our broad use of the term 'event' includes both impulsive signals and waveform changes with a less distinct onset.
We also discuss this point in a later section (line 300), and make a slight modification for consistency. Modified text: The varied nature of cryoseismicity raises the question of how an 'event' should be defined for inclusion in the catalogue.

o    Their goal is to bust a "catch-all" catalog, but there are better methods for this.
o    Tremor and all event types can be simultaneously detected and located by migration based or beam-forming methods (eg, Nanni et al GRL 2022).
o    It's not that I did not understand their goal …
o    They usually answer by referring to "part B"…
o    … But the interpretation of the results is left for a future paper.

**R1.2ndRev.02-2**
Many thanks for pressing these points. As per the note in the 'Prelude', we now better state the distinction between the 'reconnaissance' scope of our study (14 broadband stations, approx. 55 km aperture, events often recorded by less stations, median=6 stations, with some significant coupling differences being notable) and the

suggested recent 'high-resolution' work (98 Nodal-type stations, approx. 600-800 m aperture). It is exciting that our Part A + Part B workflow could help target locations within large areas that would merit a future high-resolution study, so there's certainly a nice link to the citation suggested by this review, which we are happy to include.

Last sentence of the abstract modified: The new algorithm and workflow will assist in: the comparison of different glacier environments using seismology, the identification of process change over time, and the targeting of possible following high-resolution studies.

Text modified with small additions (line 30): This workflow aims to enable the reconnaissance of ice-covered environments, such as outlet glaciers of ice sheets, some of which supply ice shelves. It provides a consistent and repeatable approach that will work with a modest number of stations deployed over a wide, remote area to provide an initial appraisal of seismicity across a given region. Such a reconnaissance could facilitate either 1) a comparison of the processes active in different locations; and/or 2) the monitoring of glacier processes over time; and/or 3) the targeting of following high-resolution studies.

Citations added (line 84): Where high-resolution sensor coverage is desirable and possible, source locations and glacier processes may be determined directly (e.g. Nanni et al., 2022 make use of a dense, ~800 m aperture array), with the reconnaissance-level approaches that we describe enabling the targeting of such detailed studies.

https://doi.org/10.1029/2021GL095996

Text modified with small additions (line 85): The wide variety of techniques for the detection of icequakes highlights the extent of analytical challenges in event-based cryoseismology. Where the area of interest is an ice stream or other ice sheet outlet glacier, the challenge is increased by the remote location together with the need to undertake a reconnaissance across a relatively large area. The diversity of event types in glacier environments therefore suggests the need for a workflow comprising…

References added for consistency (line 409): …to examine or locate a more specific event (glacier process) type (e.g. Nanni et al., 2022, Umlauft et al., 2023, Hudson et al., 2023).

https://doi.org/10.1029/2021GL095996
https://doi.org/10.1029/2023JF007280

o    For instance, the authors claimed that "Our experience is that varying the parameters mentioned results in a full catalogue". How do you know that you detected all events? Just because you detected all previously known "stick-slip"events? But there could be smaller stick-slip events that you missed, or other types of events. How to be sure?

**R1.2ndRev.02-3**
In the text of the m/s we consistently use the term 'near-comprehensive' catalogue (i.e. not 'full'). We apologise for the use of the word 'full' in our previous response. We handle the question of detected vs missed events, and also false detections, through the discussion of high confidence and low confidence events (Section 3.2.1). Given the 'catch-all' approach that we use, false detections are more of a concern than missed events, however, the general findings of the Part A m/s are not impacted in either case.

o    The multi-STA/LTA method only detect impulsive events and cannot detect tremor, which occurs frequently on glaciers.
o    The multi-STA/LTA method is also not better than other methods at detecting small events, template-matching methods can detect much smaller signals.

**R1.2ndRev.02-4**
We have expanded the examples of the multi-STA/LTA algorithm shown in Supplementary Figure S2, now S2a and S2b to illustrate the above points. As implemented in this reconnaissance for the Whillans Ice Stream (i.e. station separation approx. 10 km) the multi-STA/LTA method detects both emergent and impulsive small events quite well (Fig. S2b) and achieves what we intend, i.e. that the wide variety of events would be captured by the algorithm, e.g. contrast the large events (Fig. 2a). We do see tremor-like signals, although with the 10 km station separation, they appear in the catalogue in connection with other signals (e.g. lower plot in Fig. S2a). Text modified (line 256) to point to the updated Fig. S2a, b.

We agree that template matching is a great choice of method for high-resolution studies with a particular event type in mind (see text at lines 49-52, with small modifications), and hope that readers are inspired to

carry out well-targeted follow-up studies having carried out a reconnaissance for large glaciers aided by our Part A/Part B workflow, as in the title of the two papers.

o    I still disagree that the proposed method outperforms a simple STA/LTA algorithm. Indeed, the authors have not shown that their method is better than "any other simple STA/LTA method", just that it is better that a simple STA/lTA algorithm with extreme unrealistic values of short and long time windows.

As above, the strength of the multi-STA/LTA is that it captures a wide variety of events, and we hope that the updated Fig. S2a) and b) now provide a better sense of that.  The variety of events are then explored using the machine learning (Part B) of the workflow, by design.  Relevant text has been added in various locations as in the response to **R1.2ndRev.02-2.**

o    The main advantage of the method is that it searches automatically for the optimal values of some model parameter (time windows), but some other important parameters (triggering and detriggering thresholds) are adjusted by trial and error. The proposed method thus does not remove the need to visually optimize some model parameters.

**R1.2ndRev.02-5**

We agree with this comment, and slightly modify our existing text on this topic (line 325).

In this study we use the Monte Carlo approach to optimise the five key model parameters that have the strongest conditional interplay when applying the multi-STA/LTA method (sta, lta, $\Delta$sta and $\Delta$lta, and $\varepsilon$) as previously described (Fig. 2). Secondary parameters, which will vary based on study environment (i.e. background noise and seismic signal amplitudes) include the trigger and detrigger values. These values were set in this study following a brief, visual-based analysis as this was a straight-forward process. Whilst any parameter choices could be optimized through the Monte Carlo analysis, the needed visualization and appraisal process for the trigger values could become unwieldy. In general, the parameters that are used should be recorded and supplied with the resulting catalogue.

**c) Other points raised**

o    the tone of their manuscript and response letter is slightly pretentious and patronizing

To correct this misunderstanding, we are cautiously enthusiastic about the potential of our Part A – Part B workflow for reconnaissance of seismic signals from glaciers and ice sheets.  We aim for clarity of presentation and offer our responses with respect for other methods that are well-suited to other glacier environments..

o    For known stick-slip events, the method is not able to detect the correct start and end times (see response letter and electronic supplement).

**R1.2ndRev.03-1**

As we explain (line 206) our reconnaissance method provides a reference arrival time.  This is appropriate to the sparse recording networks that enable knowledge generation for large expanses of glaciated areas. We hope that such insights could enable subsequent high-resolution studies as in our response to R1.2ndRev.02-2.

o    Figure S2 is supposed to show a high confidence stick-slip event not previously identified.
What represent the pink areas? The time window when the event is detected at each station?
I don't see any event when looking at these seismograms. I can't believe that there is an increase in energy ratio SA/LTA larger than the chosen threshold of 3.  Or the event may be in a different frequency range than the bandpass filter 0.001-1Hz used in fig S2?  Could you show the spectrogram of this event at one or more stations? And add a plot to show the temporal evolution of STA, LTA and STA/LTA?  This event is identified as stick-slip just because it shows 3 successive pulses (electronic supplement). This is not very convincing. It could also correspond to 3 successive events. It also has a different frequency range than known stick slips events. You should at least locate this event before making a claim like that. Or show that this event repeats at regular time intervals.

**R1.2ndRev.03-2**

We have changed the presentation of Figure S2a, and figure caption, as per **R1.2ndRev.02-4.**   We didn't intend the strong claim implied by the referee's response, rather, we suggest that a different kind of slip mechanism would be plausible.

o   I understand why you filter seismograms, but why spectrograms (fig S5)?

**R1.2ndRev.03-3**

We modify the figure caption to explain that the same filter is applied to the spectrogram in this case as it relates directly to the seismogram (in this case removing a low frequency signal that overlaps with the event).

o   The authors do not filter the signal or deconvolve the data by the instrument response because it leads to "artificial detections".  I have never seen that. What kind of signals did you detect? Are these real signals (ambient noise...) or numerical problems?

**R1.2ndRev.03-4**

The most important part of our original response (we use very limited frequency-based filtering prior to applying the algorithm) is the fact that we are searching for events across a wide range of frequencies, so we need to retain that wide frequency range .  As a less important remark, a single event with different frequencies at different times sometimes appeared as separate multiple events if we presented differently pre-filtered data to the algorithm (as per 'common sense').  We withdraw the phrase 'artificial detections' as this wasn't in our original text and did not help the explanation.

We again thank Reviewer 1 for providing a second review, and hope that the above responses have now provided satisfactory explanations of the context of the work, or corrections.   It has strengthened the paper to make a better link to the high-resolution work as per the suggested citations, and we look forward to using both reconnaissance and high-resolution methods in Antarctica in the future.